# Assessing glaciogenic seeding impacts in Australia's Snowy Mountains: an ensemble modeling approach

Sisi Chen[1], Lulin Xue[1], Sarah A.  Tessendorf[1], Thomas Chubb[2], Andrew Peace[2], Suzanne Kenyon[2], Johanna Speirs[2], Jamie Wolff[1], and Bill Petzke[1]

[1]NSF National Center for Atmospheric Research (NCAR), Boulder, CO, USA
[2]Snowy Hydro Ltd., Cooma, NSW, Australia
**Correspondence:** Sisi Chen (sisichen@ucar.edu)

**Abstract.**

Winter precipitation over Australia's Snowy Mountains provide crucial water resource in the region. Cloud seeding has been operational to enhance snowfall and water storage. This study presents an ensemble simulations to assess cloud seeding impacts across diverse meteorological conditions and evaluate associated model uncertainties. Nine seeding cases from 2016 to 2019 were simulated, with 18 ensemble members varying initialization datasets and model configurations. Two main storm categories were studied (convective vs stratiform). Results demonstrate that simulated seeding efficacy highly depends on meteorological conditions. Stratiform cases exhibited consistent precipitation enhancement, while convective cases showed reductions and downwind shifts of precipitation. Significantly inter-member variability was also observed. Notably, simulations driven by the Bureau of Meteorology Atmospheric high-resolution Regional Reanalysis for Australia (BARRA) reanalysis dataset show better representation in supercooled liquid water. Aerosol and planetary boundary layer scheme variations also contributed to ensemble spread. The findings demonstrate the value of ensemble modeling for reliable cloud seeding assessment. Key areas are also identified for future investigations in winter cloud seeding.

## 1   Introduction

Glaciogenic cloud seeding has been implemented globally and in operational use for decades to enhance precipitation in winter, primarily by increasing snowfall. The process introduces artificial ice-nucleating particles (INPs), such as silver iodide (AgI), into clouds containing supercooled liquid water (SLW) to stimulate the ice nucleation. These formed ice particles can then grow and precipitate, augmenting water resources and snowpack (Flossmann et al., 2019; Manton and Warren, 2011). This method is particularly effective in winter cloud systems over mountainous terrain, where moist air is lifted and cooled, forming clouds rich in SLW. The presence of SLW in these winter orographic clouds provides an ideal environment for AgI particles to act as effective INPs at temperatures between -5 °C and -20 °C (Marcolli et al., 2016).

Winter orographic cloud seeding differs fundamentally from summertime convective cloud seeding, which targets convective clouds with strong updrafts and significant vertical development. Winter orographic seeding releases efficient INPs (normally AgI particles) into the clouds. This mode of seeding operates in more stable and predictable meteorological condi-

tions (Rasmussen et al., 2018), such as in stratiform cloud systems. Recent field campaigns, such as the Seeded and Natural Orographic Wintertime clouds—the Idaho Experiment (SNOWIE), have provided unambiguous evidence of the efficacy of winter orographic cloud seeding. Observations from SNOWIE demonstrated that AgI seeding led to enhanced snowfall rates and accumulation, confirming the fundamental winter orographic cloud seeding hypothesis (Tessendorf et al., 2019; French et al., 2018).

The Snowy Mountains in Southeast Australia offer distinct opportunities and challenges for winter orographic cloud seeding. The area serves as a vital water catchment, supporting hydroelectric power generation, irrigation, and tourism. Cloud seeding experiments were carried out during the latter half of the 20th century with generally positive results (Smith et al., 1963; Smith, 1967; Shaw and King, 1986; Warburton and Wetzel, 1992). Recognizing the importance of snowpack and potential benefits of enhancing snowfall, wintertime glaciogenic cloud seeding has been operational over the area since 2004 by Snowy Hydro Ltd. (SHL) (Huggins et al., 2008; Manton and Warren, 2011). Ground-based seeding has been implemented, releasing AgI particles from ground generators strategically located on the upwind slopes of the mountains.

Past statistical evaluations of SHL's randomized seeding program demonstrated positive impacts on precipitation over the target area, with mean precipitation increases of $\sim 0.37$ mm per seeding event during 2005–2009, and increases between 0.47 mm to 0.55 mm per event during 2010–2013 (Manton et al., 2017). Despite these promising results, assessing seeding impacts of individual events remains challenging due to significant uncertainties. In addition, the Southern Hemisphere environments are characterized by low aerosol and INP concentrations due to low levels of anthropogenic and terrestrial emissions compared to the Northern Hemisphere (Huang et al., 2017, 2021). Moreover, studies indicate that clouds over the Southern Ocean and the surrounding land masses exhibit a higher prevalence of SLW, even at colder temperatures, compared to their Northern Hemisphere counterparts (Morrison et al., 2011; Hu et al., 2010). These unique conditions enhance the seeding potential, as more liquid water is available for conversion into ice crystals with less competition from pre-existing natural INPs. However, the unique conditions in the Snowy Mountains also introduce potential challenges for modelling precipitation processes and the seeding impact compared to Northern Hemisphere environments. For example, there are uncertainties in respect to cloud phase-partitioning, riming, and collision-coalescence processes with lower atmospheric pollution, and most cloud microphysical parameterizations are developed and validated based on the observations made in the Northern Hemisphere (e.g.,Meyers et al. (1992); Thompson et al. (2008); Thompson and Eidhammer (2014); Eidhammer et al. (2009). These differences may affect model performance on representing clouds and precipitation processes in the Snowy Mountains. Addressing these uncertainties and challenges requires detailed numerical modeling to simulate cloud seeding under various atmospheric conditions and systematic and comprehensive assessment of the impacts.

Previous work by Chen et al. (2023) used the WRF-WxMod® model to simulate cloud seeding over the Snowy Mountains. The study focused on three seeding cases during different synoptic weather events in the 2018 winter season, and demonstrated that WRF-WxMod® can realistically represent cloud structures, liquid water path (LWP,) and precipitation patterns and amount in both natural and seeded scenarios. Sensitivity analyses revealed significant variability in model responses due to factors such as aerosol concentrations, ice nucleation efficiencies, and initialization datasets. The study also indicates a weaker model sensitivity to the secondary ice production efficiency due to Hallett-Mossop (HM) processes. Modifying the efficiency only imposed

negligible impacts on clouds and precipitation compared to the effect of changing the IN schemes. This is most likely due to the fact that clouds contained a large amount of supercooled liquid but very little graupel, which is not enough to activate the secondary ice production processes. Importantly, no single model configuration optimally represented all cases, with complex interaction between seeding particles, clouds, and the large-scale meteorological conditions.These findings highlighted the necessity of employing an ensemble modeling approach to more comprehensively assess seeding impacts and capture the range of uncertainties inherent in the simulations. An ensemble approach allows for the exploration of multiple model configurations and input datasets, providing a probabilistic framework that can account for variations in initial conditions, physical parameterizations, and other key factors influencing model outputs.

Building upon the work by Chen et al. (2023), this study aims to develop an ensemble modeling framework to more systematically and comprehensively evaluate the impacts of glaciogenic cloud seeding in different winter weather regimes over the Snowy Mountains. Ensemble studies by Xue et al. (2022) for SNOWIE and Rasmussen et al. (2018) for the Wyoming Weather Modification Pilot Project (WWMPP) have demonstrated the effectiveness of this approach in simulating and quantifying the seeding impacts in North America. Their studies highlight the importance of ensemble methods in capturing the variability of seeding effects due to differences in initialization data, physical parameterizations, and atmospheric conditions. By adapting this approach and tuning it to be better suitable for wintertime conditions in the Snowy Mountains, we aim to provide a reliable, comprehensive, and systematic assessment of the cloud seeding impacts and knowledge of the model variability and uncertainty over the Southern Hemisphere environment. This work seeks to address the following objectives:

1. Evaluate the variability of seeding impacts, as simulated by WRF-WxMod ®, across different winter cloud conditions in Snowy mountains.

2. Assess the model sensitivities to different configurations in those conditions.

3. Test the robustness of the Bureau of Meteorology Atmospheric high-resolution Regional Reanalysis for Australia, version 1 (Su et al., 2019, BARRA) as a new initialization dataset to drive WRF simulations.

4. Determine a tailored ensemble modeling framework for the region by incorporating a diverse set of appropriate model configurations and initialization datasets.

The paper is outlined as follows. The Model and Data Section will introduce the WRF-WxMod® model used for conducting simulations of the seeded and/or natural cloud processes, the case selections over the past decades, and observational datasets used for model validation. Ensemble simulation analysis of the natural precipitation and seeding impacts will be presented in the Results and Discussion section, followed by Summary.

## 2 Model and Data

### 2.1 The WRF-WxMod Cloud Seeding Model

WRF-WxMod® (pronounced "WRF weather mod") model extends the capabilities of the Weather Research and Forecasting (WRF) model (Skamarock et al., 2008) to simulate the microphysical processes associated with AgI seeding (Xue et al., 2013a, b). The model incorporates microphysical parameterizations for the release, dispersion, and nucleation of AgI particles,

accounting for four ice nucleation mechanisms (deposition, condensation freezing, contact freezing, and immersion freezing) following the formulae of DeMott et al. (2010) and Meyers et al. (1992). AgI particles are treated as a single-mode lognormal distribution (with a geometric mean diameter of 40 nm and a geometric standard deviation of 2.0) and can act as INPs or cloud condensation nuclei (CCN) due to their soluble components. The AgI particles can be scavenged by liquid drops and ice crystals through processes such as Brownian diffusion, turbulent diffusion, and phoretic effects (thermophoresis and diffusiophoresis), which subsequently determines the AgI nucleation through immersion freezing and contact freezing. The detailed description and formulation can be found in Xue et al. (2013a). The model is capable of simulating seeding released from ground-based generators, ejectable flares and burn-in-place at cloud tops by aircrafts. Once released into the atmosphere, the AgI particles are transported by winds. As they disperse, they interact with existing cloud hydrometeors through various microphysical processes. These interactions facilitate the nucleation and growth of ice crystals, ultimately leading to the formation of different types of precipitating hydrometeors such as snow and graupel. Therefore, the model allows for a detailed investigation and quantification of how AgI seeding influences cloud microphysics and precipitation development.

In this study, we focus on ground-based seeding, as no airborne seeding operations were conducted over the area. Ground-based AgI generators are modeled as point sources positioned at operational sites within the simulation domain. The AgI emission rate was prescribed based on the actual generator operations in Snowy mountains, approximately 20.6 g hr$^{-1}$, close to the emission rate (= 20.4 g hr$^{-1}$) reported in Huggins et al. (2008). The model domain comprises two nested grids: 4-km outer domain encompassing Southeast Australia to capture the large-scale synoptic weather patterns and a 1-km inner domain focused on the Snowy Mountains catchment (Figure 1). The large outer domain allows for adequate spin-up of the upper wind area under the prevailing westerly winds in wintertime SE Australia. The finer resolution is critical for resolving the orographic effects and detailed cloud microphysical processes, and therefore the ensemble members only comprise the 1-km domain simulations.

## 2.2 Ensemble design

Building upon the sensitivity study of cloud seeding simulations over Snowy Mountains by Chen et al. (2023), we carefully selected the model configurations for our ensemble members to address the key model sensitivities identified in previous simulations to estimate the ensemble spread across different cases.

### 2.2.1 Initialization Datasets

In Chen et al. (2023), two different reanalysis datasets were used to drive the simulation: European Centre for Medium-Range Weather Forecasts (ECMWF) Reanalysis v5 (ERA5, Hersbach et al., 2020) and National Centers for Environmental Prediction's (NCEP's) Climate Forecast System, version 2 (CFSFv2, Saha et al., 2014). It was found that the choice of initialization dataset contributed to the largest uncertainty in the simulations. Different reanalysis datasets provide varying large-scale atmospheric forcings due to differences in data assimilation methods and model physics. In this study we introduced the third initialization dataset, the 6-hourly Bureau of Meteorology Atmospheric high-resolution Regional Reanalysis for Australia, ver-

sion 1 (BARRA, Su et al., 2019) to improve the ability to discern relative strengths and weaknesses among the initialization datasets and enhance the representativeness of the ensemble members.

BARRA offers a comprehensive suite of gridded meteorological datasets covering Australia, New Zealand, and a significant expanse of SE Asian countries. The BARRA dataset boasts a higher spatial resolution compared to the commonly used global reanalysis dataset such as ERA5 ($\sim$ 31 km) and CFSv2 ($\sim$ 38 km), by providing a 12-km resolution whole-domain dataset (BARRA-R) and four additional convective-scale (1.5-km) nested domains centered on major Australian cities (BARRA-C, BARRA-PH, BARRA-AD, and BARRA-TA, Su et al. 2021). In this study, the 12-km BARRA-R, which covered the entire continental Australia and the surrounding oceans, was used. Such enhanced spatial resolution allows BARRA to provide more detailed initial and boundary conditions for numerical simulation. Su et al. (2019) shows that BARRA-R shows reduced errors in 2 m temperature, 10 m wind speed, and surface pressure observations, compared to global reanalysis ERA-Interim and MERRA-2.

This is the first time to our knowledge that BARRA has been used to drive WRF simulations. As mentioned in the introduction, one of our scientific objectives is to demonstrate whether BARRA can serve as a reliable initialization dataset for WRF for simulating the winter orographic clouds and precipitation over the Snowy Mountains of Australia. Comparisons of simulations driven by ERA5, CFSv2, and BARRA-R against observations were conducted to evaluate the performance of each dataset.

### 2.2.2 Planetary Boundary Layer Schemes

Chen et al. (2023) only considered one planetary boundary layer (PBL) scheme in their study. However, their study indicated that seeding impacts can vary a lot under different atmospheric conditions, particularly in relation to atmospheric stability and wind profiles. This indicates that the seeding effects are sensitive to planetary boundary layer (PBL) processes, which not only influence turbulence, mixing, and the development of clouds, but also affects how seeding particles are transported and dispersed by the winds. Therefore, in the current study, four PBL schemes were used to encompass the uncertainties caused by PBL schemes: Mellor-Yamada Nakanishi and Niino Level 2.5 (MYNN, Nakanishi and Niino, 2009), Mellor-Yamada-Janjic (MYJ, Janjić, 1990), Quasi-Normal Scale Elimination (QNSE, Sukoriansky et al., 2005), and Yonsei University (YSU, Hong et al., 2006).

### 2.2.3 Aerosol background

Both monthly aerosol climatology and reduced CCN (pristine environment) will be included in the ensemble members. Chen et al. (2023) found that reducing the aerosol concentrations to 10% of the monthly climatology (denoted as 01CCN) in Thompson-Eidhammer scheme (Thompson and Eidhammer, 2014) enhanced warm rain processes in the natural (no seed) conditions. This enhancement led to a stronger phase conversion from liquid precipitation to ice precipitation in the seeded simulations. Since ice-phase hydrometeors, such as snow and graupel, have slower fall speeds compared to raindrops, this resulted in precipitation falling further downstream, effectively shifting the precipitation distribution. Due to a lack of in situ measurements to verify the CCN, droplets, and ice concentration, two CCN concentrations (monthly climatology and 10% of climatology) were considered to address the model uncertainty.

### 2.2.4 Ice nucleation schemes

Two ice nucleation schemes were considered (DeMott et al., 2010; Meyers et al., 1992, hereafter referred to as DeMott and Meyers Scheme, respectively) to take into account the uncertainties in the nucleated ice in the natural environment. Even though the seeding impacts on the precipitation amount reaching the ground remained similar over the target area, Chen et al. (2023) found that changing the ice nucleation scheme effectively altered the amounts of liquid(ice)-phase precipitation reduction (increase) due to seeding. This suggests that, although the overall effects on precipitation change might be less sensitive, the microphysical processes are sensitive to the choice of ice nucleation parameterization. Including both schemes enabled us to examine how different representations and efficiencies of ice nucleation processes influence seeding impacts.

### 2.2.5 Summary of ensemble configurations

By carefully considering the extent of uncertainties to which the above-mentioned key parameters generated, we finalized 18 ensemble members to systematically explore these sensitivities, aiming to capture a comprehensive range of uncertainties inherent in modeling cloud seeding impacts. Each ensemble member represents a unique combination of aerosol conditions, ice nucleation processes, boundary layer dynamics, and initialization datasets. For each member, we conducted both SEED and NO SEED (control) simulations, allowing for direct comparison and quantification of seeding impacts on cloud structure, cloud/precipitation process rates, and the distribution and amount of precipitation at the surface. This approach enables us to assess the robustness of simulated seeding impacts under different winter environments and to identify conditions under which the model is most sensitive to specific parameters. The physics schemes and configurations considered in the ensemble members, expanded upon the case studies of Chen et al. (2023), summarized in Table 1. At the time the study was conducted, the BARRA reanalysis data covered the dates from January 1, 1990 through February 28, 2019, so cases after February 28, 2019 only consist of ERA5 and CFSv2 members.

### 2.3 Case Selection and Observational data

To systematically analyze the seeding impact, we classified the meteorological conditions of all past seeding Experimental Units (EUs) from the 2016-2019 seeding seasons into two main groups: one with an unstable environment associated with convective, deep clouds, high wind, and high precipitation (referred to as category 0), the other with a relatively stable environment associated with stratiform orographic clouds, calm wind, and weak precipitation (referred to as category 1). An EU is a 5 hour period during which a randomized seeding experiment was executed. Nine EUs were selected for this study: five in category 0 and four in category 1, allowing for a comprehensive evaluation of cloud seeding impacts under both conditions. Table 2 summarizes the nine EUs. For clarity, we will use case numbers (column one in Table 2) grouped by categories instead of EU numbers in the following sections. Our goal was to capture the variability in key cloud and precipitation properties through the ensemble approach and provide a robust assessment of seeding impacts in two distinct winter conditions.

To validate the model and evaluate the ensemble results, observational data collected during the seeding periods were used, including atmospheric soundings, radiometer measurements and precipitation from a network of all weather gauges. Table 3

**Table 1.** List of 18 ensemble configurations for each case. For each configuration, one control (no seed) simulation and one seed simulation were conducted. In total 18 control members and 18 seed members were run. The BARRA reanalysis data is only available up until 2019-02-28. Therefore, cases after this date only consist of ERA5 and CFSv2 members (i.e., 12 members). The changed parameters are highlighted in bold within each initialization dataset group, relative to the "default configuration" (CCN_DeMott_MYNN).

| Ensemble number | Ensemble configuration name | Initialization dataset | CCN concentration | IN scheme | PBL scheme |
|---|---|---|---|---|---|
| 1 | ERA5_CCN_DeMott_MYNN | | Monthly climatology | DeMott | MYNN |
| 2 | ERA5_CCN_DeMott_MYJ | | Monthly climatology | DeMott | **MYJ** |
| 3 | ERA5_CCN_DeMott_QNSE | ERA5 | Monthly climatology | DeMott | **QNSE** |
| 4 | ERA5_CCN_DeMott_YSU | | Monthly climatology | DeMott | **YSU** |
| 5 | ERA5_01CCN_DeMott_MYNN | | **10% climatology** | DeMott | MYNN |
| 6 | ERA5_CCN_Meyers_MYNN | | Monthly climatology | **Meyers** | MYNN |
| 7 | CFS2_CCN_DeMott_MYNN | | Monthly climatology | DeMott | MYNN |
| 8 | CFS2_CCN_DeMott_MYJ | | Monthly climatology | DeMott | **MYJ** |
| 9 | CFS2_CCN_DeMott_QNSE | CFSv2 | Monthly climatology | DeMott | **QNSE** |
| 10 | CFS2_CCN_DeMott_YSU | | Monthly climatology | DeMott | **YSU** |
| 11 | CFS2_01CCN_DeMott_MYNN | | **10% climatology** | DeMott | MYNN |
| 12 | CFS2_CCN_Meyers_MYNN | | Monthly climatology | **Meyers** | MYNN |
| 13 | BARRA_CCN_DeMott_MYNN | | Monthly climatology | DeMott | MYNN |
| 14 | BARRA_CCN_DeMott_MYJ | | Monthly climatology | DeMott | **MYJ** |
| 15 | BARRA_CCN_DeMott_QNSE | BARRA | Monthly climatology | DeMott | **QNSE** |
| 16 | BARRA_CCN_DeMott_YSU | | Monthly climatology | DeMott | **YSU** |
| 17 | BARRA_01CCN_DeMott_MYNN | | **10% climatology** | DeMott | MYNN |
| 18 | BARRA_CCN_Meyers_MYNN | | Monthly climatology | **Meyers** | MYNN |

summarizes the observational datasets, and locations of instrumentation are shown in Figure 1. Atmospheric soundings provide
information on the vertical structure of the atmosphere including atmospheric stability and cloud depth. LWP measurement is a good proxy for SLW in winter clouds when most of the clouds are above freezing levels, and SLW is not only critical for natural ice processes, such as ice production and riming, but also a necessary condition for seeding particles to take effect. Therefore, accurate representation of the available SLW in the model in terms of amount and trend is essential for simulating the microphysical processes in natural and seeded environments. By comparing the model's LWP outputs with radiometer
observations, we can assess the model's performance in capturing the key microphysical characteristics of the clouds over the Snowy Mountains. Accumulated precipitation from gauges are used to verify how the model resolves the distribution and amount of precipitation on ground.

**Table 2.** Summary of the nine cases selected for simulations. Whether the case was seeded during the experimental unit (EU) period in the actual operations is indicated in the fourth column. For clarity, the case number, instead of EU number, will be referred for the rest of the paper. The case is named by the cloud category (0 for convective, 1 for orographic stratiform) followed by a unique index within that category.

| Case | EU | Meteorological conditions | Seeding action | Simulated time for analysis (UTC) |
|---|---|---|---|---|
| 0a | EU258 | | No seed | 2016-06-23 18:00 – 06-24 02:00 |
| 0b | EU318 | | Seed | 2018-08-06 00:00 – 08-06 12:00 |
| 0c | EU319 | Windy, high precipitation, convective and deep clouds | No seed | 2018-08-06 09:00 – 08-06 17:00 |
| 0d | EU331 | | Seed | 2019-05-28 20:00 – 05-29 04:00 |
| 0e | EU349 | | No seed | 2019-08-09 06:45 – 08-09 15:00 |
| 1a | EU274 | | No seed | 2016-08-20 05:00 – 08-20 13:00 |
| 1b | EU326 | Low wind, weak precipitation, stratiform orographic clouds | No seed | 2018-08-18 13:30 – 08-18 22:00 |
| 1c | EU327 | | Seed | 2018-08-20 14:00 – 08-21 00:00 |
| 1d | EU352 | | No seed | 2019-08-10 07:00 – 08-10 12:30 |

**Table 3.** List of ground-based observational data used for model-observation comparisons. Locations of the sites are shown in Figure 1.

| Measurements | Description | Locations |
|---|---|---|
| Sounding | Radiosondes suspended below weather balloons, provide ~3 hourly vertical profiles of temperature, relative humidity, wind speed, and wind direction. | Khancoban site |
| Radiometer | Vertically integrated liquid water path (LWP) | Cabramurra site |
| All weather precipitation gauge | Mix of NOAH-II all weather precipitation gauges and heated/unheated (depending on elevation) tipping bucket gauges. Real-time "tips" recorded and post-processed to 10 minutes. | 67 sites operated by Snowy Hydro in the Snowy Mountains |

# 3  Simulated results and discussion

## 3.1  Comparison with Observations

### 3.1.1  Liquid Water Path

The LWP data, representing vertically integrated liquid water content, was collected from the radiometer measurement at the Cabramurra site (1500 m above sea level). The radiometer generally lies above freezing level ($< 0$ °C) for most cases, which

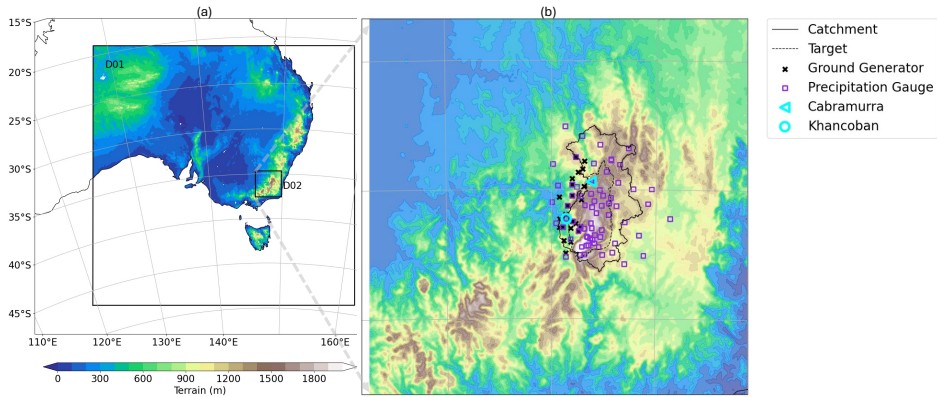

**Figure 1.** (a) Terrain map of the nested model domains for the ensemble simulations. The 4-km outer domain covers the large area of Southeast Australia, the 1-km inner domain covers the Snowy Hydro water catchment (outlined in black solid line). (b) The zoomed-in terrain map of the inner domain. The ground generators (black x's) are located on the western slope of the catchment. The target area within the catchment (dashed black line) is the region where seeding impacts on precipitation were expected across the domain. The precipitation gauges are shown in blue violet squares. The radiometer data was collected at the Cabramurra site (large cyan triangle), and atmospheric sounding data was collected at the Khancoban site (large cyan circle).

makes the LWP measurements a good proxy for the presence of SLW in clouds. SLW is critical for both natural ice production and the effectiveness of glaciogenic seeding, which converts supercooled liquid to ice. Therefore an accurate representation of

205 SLW in the model is critical to accurately capture both natural and seeded clouds. Observations reveal that winter orographic clouds in the Snowy Mountains contain abundant SLW, with LWP values generally between $0.5 - 1.5$ kg m$^{-2}$ (Figure 2), significantly higher than values measured in the Wyoming Weather Modification Pilot Project (WWMPP, Rasmussen et al., 2018) which were generally lower than $0.2$ kg m$^{-2}$ and SNOWIE campaign (Xue et al., 2022) which were generally lower than $0.6$ kg m$^{-2}$.

Model simulations captured the range of variation of LWP in four cases (Cases 0a, 0b, 1a, 1b) (Figure 2). However, discrepancies were noted in other cases, with simulations slightly underestimating (Cases 0c, 0e, 0d, 1d) or overestimating (Case 1c) LWP in specific time periods. These challenges highlight the inherent difficulties in accurately simulating SLW despite considering multiple model configurations. A highly variable LWP can be observed from the radiometer (e.g., Case 0c), while the model tends to produce a less variable LWP. This could be due to the factor that a 1-km grid may not capture the temporal

variability originated from subgrid scale impacts at the single-point station site. Nevertheless, the model was able to capture the high SLW environments in the Snowy Mountains.

     Simulations driven by the high-resolution BARRA reanalysis data generally outperform ERA5 and CFSv2 in most cases, producing the best representation of LWP. For example in Cases 0b and 1b (Figure 2b,h), CFSv2 and ERA5 produced a significantly larger LWP bias at the beginning of the simulation, while the BARRA-driven members almost reproduced the observed

trend, implying that BARRA provides a better initial and boundary conditions for advecting and producing supercooled liquid.

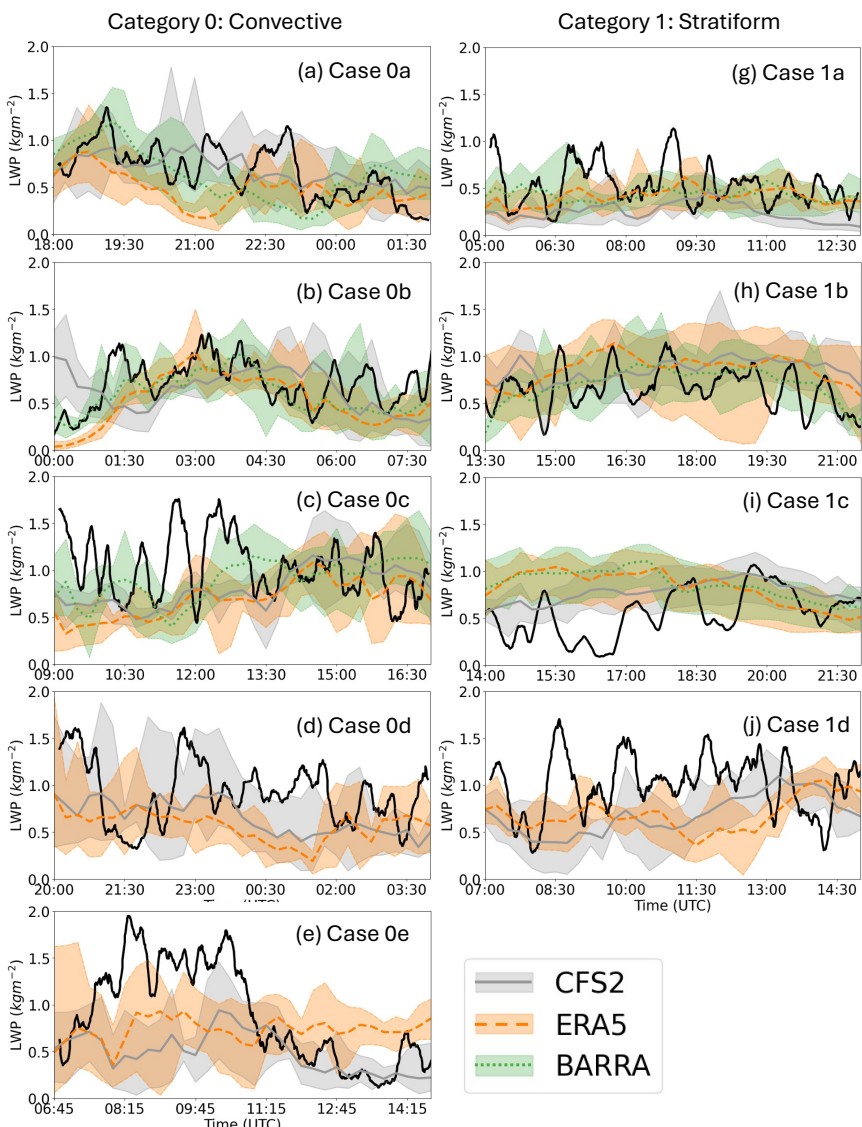

**Figure 2.** Time series of observed LWP (solid thick black lines) and simulated LWP (colored lines) at Cabramurra. The cases from the convective category are shown in the left column (a-e) and stratiform category in the right column (g-j). Individual model members are not shown for simplicity, and rather we emphasize the ensemble spread contributed by different initialization datasets. The shaded colors indicate the value ranges from the three initialization dataset member groups (six members from each group), with the ensemble mean shown in colored lines.

One exception was Case 1c where both BARRA and ERA5 members produced moderately higher LWP in the first half of the simulation, and the CFSv2 members performed better (Figure 2i). In cases where BARRA data were not available, performance of CFSv2 and ERA5 were comparable with no dominant one outperforming the other.

Variations in CCN concentrations, PBL schemes, and IN schemes all contributed to ensemble spread but did not show consistent patterns (not shown). Among these factors, the initialization datasets exerted the most significant influence on LWP variability, emphasizing the importance of large-scale forcing in accurately capturing SLW.

### 3.1.2 Precipitation

Precipitation gauge observations across the Snowy Mountains water catchment provided a crucial dataset for evaluating the model's ability to replicate the spatial distribution, magnitude, and trends of precipitation as a result of complex dynamical, thermodynamical, and microphysical processes across scales. The ensemble spread of gauge site-averaged accumulated precipitation displays a good balance of member selection (i.e. an even spread by different configurations). Overall, the model captured the high precipitation trend in Category 0 cases and weak precipitation trend in Category 1 cases (Figure 3). The ensemble spread encompassed observed values in five cases (Cases 0b, 0c, 0d, 1a, and 1d), but the remaining four cases (Cases 0a, 0e, 1b, 1c) show that all members produce higher precipitation than observation.

In most cases, the ensemble spread contributed by varying different model configurations does not show a universal bias pattern, and the dominant factors are case-specific. Precipitation spread in Cases 0a and 0e was largely driven by the choice of initialization datasets, as the spread of accumulated precipitation is clustered around each initialization datasets (Figure 3a). Nevertheless, no consistent bias pattern was found as to which initialization dataset(s) produced the most realistic precipitation. For example, in Case 0a, CFSv2 members produced higher precipitation and ERA5 members lower. The opposite trend was found in Case 0e. It was also found that different initialization datasets produced a very similar range in ensemble spread in Category 1 (stratiform) cases.

In contrast, aerosol concentration contributed most to spread in Cases 0b, 0c, 1a, and 1c. In all four cases, reduced CCN conditions generally produced the highest precipitation among all members (Figure 3c). However, in cases where aerosol conditions are not the leading contributor, reducing CCN did not result in highest precipitation.

For Cases 0d, 1b, and 1c, PBL schemes had the greatest influence on the ensemble spread, though their influence is not consistent across cases. For example in Cases 0d and 1b, QNSE members produced the highest precipitation of all, and MYNN produced the least precipitation. However, Case 1c shows the opposite trend. And in Case 0c, MYJ produced the lowest precipitation while the difference from the other three schemes is small. In Cases 0e and 1a, the four PBL schemes produced a very similar spread.

The accumulated precipitation distribution from the control (no seed) simulations of both categories shows consistent spatial patterns, with maximum precipitation along the high terrain in the target area (Figure 1b) and minimal precipitation in upwind and downwind regions. This pattern was consistent with the precipitation gauge observations (Figure 4). Standard deviation maps of the ensemble members also revealed a higher variation over the areas of maximum precipitation (right column in

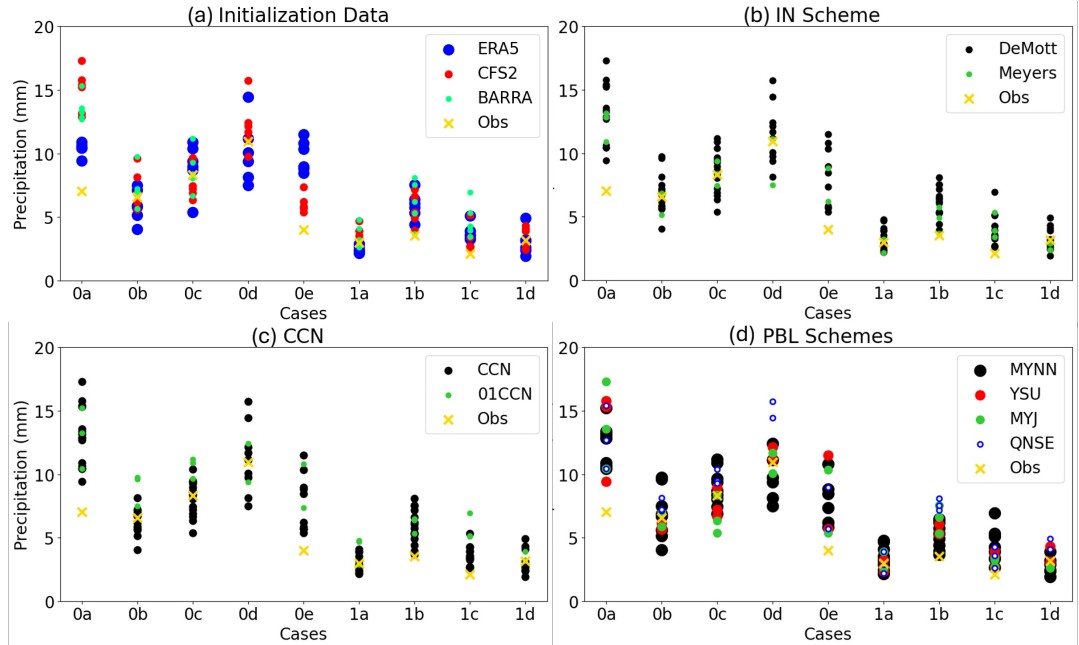

**Figure 3.** Exploded box plot of the ensemble spread of the accumulated precipitation averaged over the precipitation gauge sites for all cases from both categories. The observed gauge-site average accumulated precipitation for the same period is shown in golden "x". Each column in the plot denotes the spread of one case. Panels (a-d) show the ensemble spread contributed by varying four different model configurations: (a) initialization datasets, (b) IN schemes, (c) CCN concentrations, and (d) PBL schemes. Colors and symbols differentiate the model configurations.

Figure 4 ). Nevertheless, the coefficient of variation in those areas was smaller than 1, i.e., the standard deviation is smaller
than the ensemble mean, indicating that the ensemble spread was well constrained.

To further quantify how each model configuration contributes to ensemble spread, we calculated the difference between a fixed default ensemble member and that of a variation member. The default configuration is defined as "CCN_DeMott_MYNN", present under each initialization dataset group (as highlighted in Table 1). The variation members always have only one changed parameter (CCN, IN scheme, or PBL scheme) relative to the default member, across the three initialization datasets. There-
fore, in each case, we have 6 members for CCN sensitivity (2 CCN concentrations × 3 initialization datasets), 6 members for IN sensitivity (2 IN schemes × 3 initialization datasets), 12 members for PBL sensitivity (4 PBL schemes × 3 initialization datasets), and 18 members for Initialization dataset sensitivity (3 variations in initialization dataset × 6 members within each dataset). Therefore the difference between the variation members and default member within one configuration group provides ensemble difference attributed to changes of that configuration.
Figure 5 summarizes the ensemble differences attributed to the four configurations, with results stratified by storm category. PBL scheme and initialization dataset emerged as the largest contributors to domain-averaged ensemble spread in Category 0

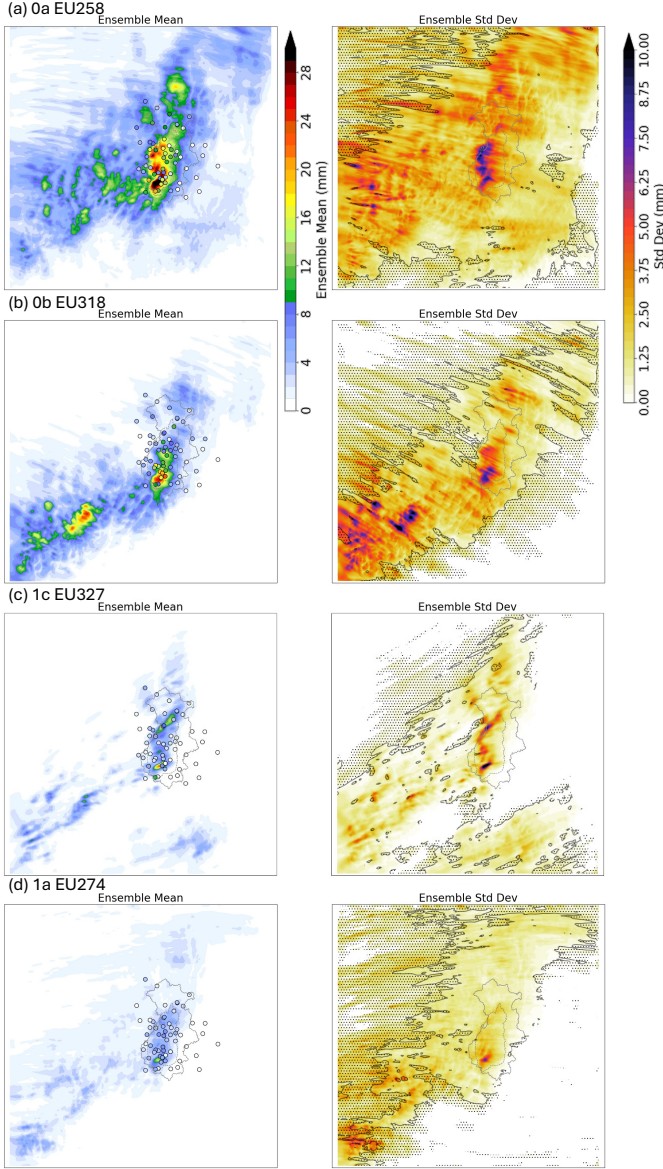

**Figure 4.** (Left column) ensemble mean of the accumulated precipitation field and (right column) the standard deviation of the ensemble members. The top two panels are from Category 0 (convective and high precipitation) and the bottom two panels are from Category 1 (stratiform and weak precipitation). From each category, one case with higher gauge-site average precipitation than observations (Panel (a): Case 0a; Panel (c) Case 1c) and one case with a comparable gauge-site average precipitation to observations(Panel (b) Case 0b; Panel (d) Case 1a) are selected, respectively. The color-filled circles indicate the observation from precipitation gauges. The hatched area in the standard deviation map indicates coefficient of variation (CV) > 1 (i.e., standard deviation > mean), and the area encircled by solid black line is the region with CV ≤ 1.

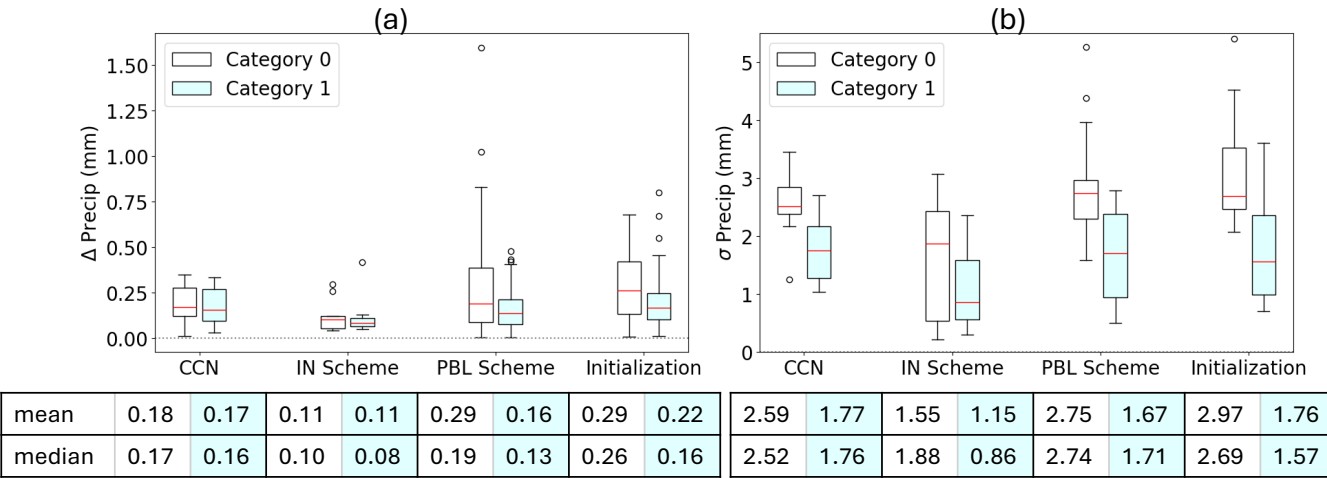

**Figure 5.** (a) Domain-average and (b) standard deviation of spatial distribution of ensemble difference in accumulated precipitation (variation member - default member, in mm) over the 1 km domain. Ensemble differences resulting from variations in CCN concentration, IN schemes, PBL schemes, and initialization datasets are shown separately. Results are stratified by storm category: Category 0 (white bars) and Category 1 (light cyan bars) appear side by side. In panel (a), absolute values of the domain-averaged differences are used to highlight magnitude and avoid cancellation effects around zero. Mean and median values for each storm category are shown in the tables beneath each variation group.

(convective) cases, both with a mean difference of 0.29 mm. In Category 1 (stratiform) cases, uncetainties attributed to PBL and initialization datasets reduced to that comparable to CCN. And consistent with the findings from Figure 3, IN schemes showed the weakest model sensitivity, regardless of storm category. Interestingly, for the spatial variability, convective storms clearly show a significantly higher sensitivity across all four configurations. In both storm categories, PBL, Initialization datasets, and CCN contribute equally to the ensemble differences, while IN schemes showed the lowest sensitivity.

### 3.1.3 Cloud Vertical Structure and Atmospheric Stability

Sounding data from the Khancoban site (Figure 1) provided profiles of temperature, humidity, and wind to evaluate the model's ability to replicate cloud vertical structure (e.g., cloud top height, cloud depth) and atmospheric stability (e.g., inversion layer and wind shear). The ensemble results demonstrate good agreement with the observed temperature profiles, instability layers, wind profiles, cloud depth, and cloud height (Figure 6). For Category 0 cases, the model captured the deeply convective cloud layer and unstable atmospheric conditions. For example, in Cases 0a and 0b (Figure 6a-b), a constant equivalent potential temperature ($\theta_e$) and saturated relative humidity (RH) up to 6 km is seen in both observed and simulated profiles, indicating a well-mixed layer and a deep cloud layer topped at 6 km. For Category 1 cases, the model produced shallow clouds and a stable atmosphere. For example both Cases 1c and 1a show shallow cloud top around 3-4 km from the RH profile, and a negative lapse rate in e (Figure 6 c-d). Overall, the model tends to produce a smoother profile in temperature and wind than observations.

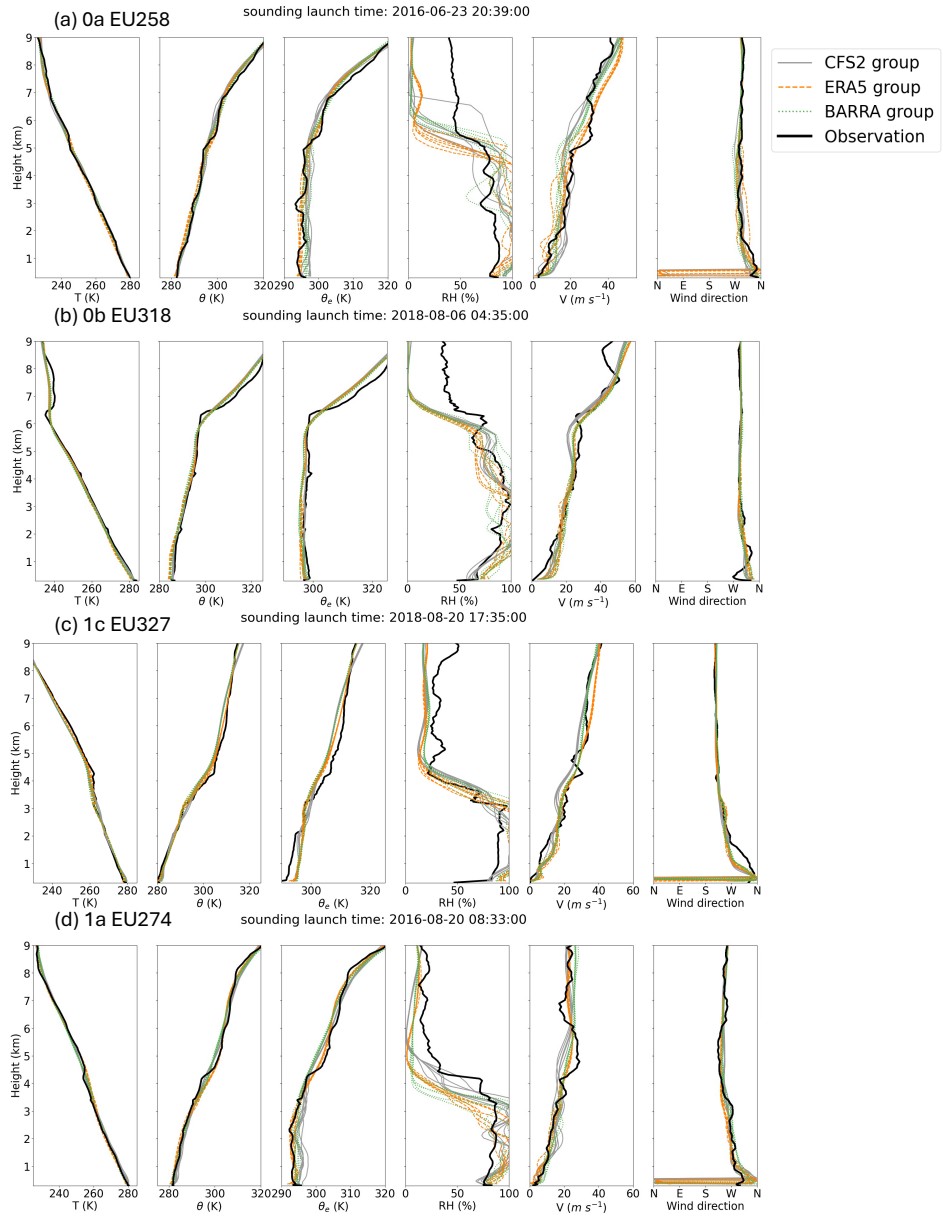

**Figure 6.** Observed (thick, solid, black lines) and simulated (thin, colored lines) sounding profiles at the Khancoban site during the four cases same as Figure. 4: (a) Case 0a with sounding launched at 20:39:00, June 23, 2016 and simulated sounding valid at 20:30:00, June 23, 2016, (b) Case 0b with sounding launched at 04:35:00, Aug 6, 2018 and simulated sounding valid at 04:30:00, Aug 6, 2018, (c) Case 1c with sounding launched at 17:35:00 Aug 20, 2018 and simulated sounding validated at 17:30:00, Aug 20, 2018, and (d) Case 1a with sounding launched at 08:33:00, Aug 20, 2018 and simulated sounding validated at 08:30:00, Aug 20, 2018.

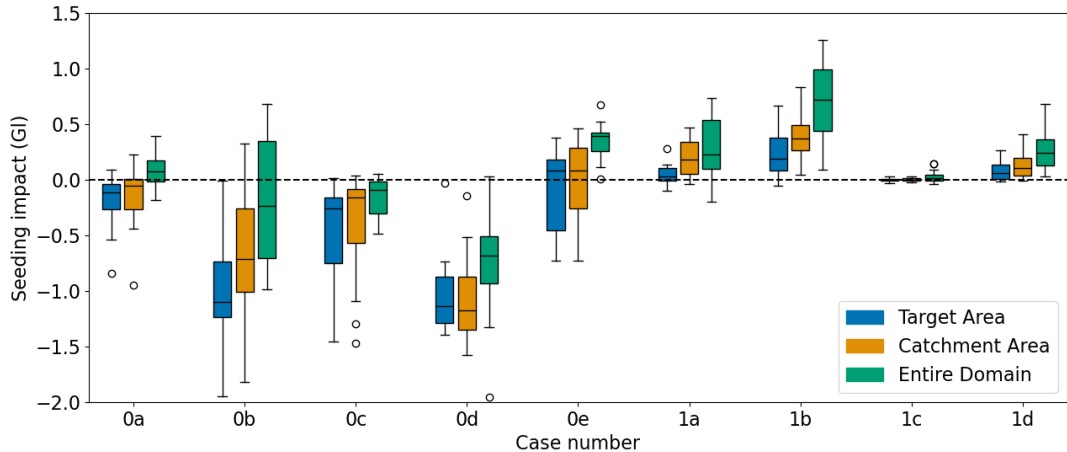

**Figure 7.** Box plots showing ensemble spread of the simulated precipitation changes in volume (in gigalitres (Gl), 1 Gl = $1 \times 10^6$ m$^3$) due to seeding (Seeded - Non-seeded) over the target area (blue), water catchment (orange), and entire 1 km domain (green) for all cases from category 0 (left) and category 1 (right).

The RH profiles exhibited significant variability and large deviation between model and observations, partly due to the RH sensor's dry bias within cloud layers. It was pointed out by Chen et al. (2023) that the RH sensor from DFM-09 sondes underestimates RH near saturation, which was a well-known issue for some other RH sensors as well (Vaisala RH sensor, Vömel et al., 2007) These sensor biases also influenced the equivalent potential temperature ($_e$) as it was derived from both temperature and humidity profiles, contributing to low observational biases in the lower level when humidity is approaching saturation (e.g., Figure 6a,c and d).

### 3.2 Ensemble Analysis of Simulated Seeding Effects

To evaluate the impacts of seeding across various model configurations, paired control (no seed) and seeding simulations were conducted for each ensemble configuration. The differences between the two simulations can then be utilized for quantification and spatial mapping of simulated seeding effects.

#### 3.2.1 Precipitation Responses to Seeding

Figure 7 illustrates the ensemble spread of simulated precipitation changes due to seeding across target, catchment, and model domain area. Except for Case 1c showing weak seeding impacts, all cases exhibited strong inter-member variabilities, with larger inter-member variability observed in Category 0 (convective, deep clouds) cases compared to Category 1 (stratiform, shallow clouds) cases.

Clear, opposite seeding impacts emerged between the two weather categories: category 1 cases consistently showed positive median seeding impacts across ensemble members, whereas category 0 cases exhibited negative median impacts within the

target area, except for Case 0e. Notably, in Case 0e, the northern target area experienced precipitation reductions, while seeding the southern region saw increases, resulting in an overall positive median value (Figure 7). Idealized cloud seeding simulations by Geresdi et al. (2017) also found negative seeding impacts in most convective clouds and positive seeding in layered clouds (Figure 10 in their paper). The negative impact in convective clouds was attributed to the weakened riming process as a result of an increased ice concentration and significantly reduced available SLW for riming. Our simulations also show a more significant reduction in liquid-phase precipitation over the catchment area due to seeding in Category 0 cases than in Category 1 cases (Figure 8c, f), which results in a significant reduction in the rimed water. The enhancement in ice-phase precipitation does not compensate for the reduction in liquid precipitation in the convective cases, resulting in an overall negative change in total precipitation reaching ground.

Model results show that the seeding impacted area can extend beyond the target area. Higher water volume gain or less water volume loss was simulated when downwind areas outside the target were included (Figure 7). Seeding resulted in a discernible downwind shift in precipitation (Figure 8a). This shift was associated with the active phase transformation of SLW into ice, which reduced liquid-phase precipitation and increased ice-phase precipitation (Figure 8b, c). Slower fall speeds of ice hydrometeors facilitate their transport further downwind, causing increased precipitation to the east of the target area and slight reductions in the upper wind area within the target area. This downwind shift in precipitation was more pronounced in Category 0 due to high winds and convective conditions, as can be seen in the less negative seeding impacts when including more downwind regions. In particular, Case 0a showed the interquartile range of seeding impacts shifted from negative over target area to positive over the 900m entire domain. The downwind shift was also present in Category 1 but to a lesser extent (see contrasts between Figures 8a and 8d). Therefore positive median values were observed in all three domains for all the Category 1 cases (Figure 7).

In addition, this downwind shift can also be complicated by the growth mechanism of ice hydrometeors. As snow falls slower than graupel, how seeding influences the riming processes, and how the model represents the riming efficiency also determines the partitioning of snow and graupel, which eventually affects the location of the precipitation that falls on the ground. Therefore, future studies regarding the sensitivity of the hydrometeor partitioning (snow vs graupel) and seeding impacts to the riming efficiency are needed to comprehend the model uncertainties. Furthermore, the convective, unstable conditions in Category 0 cases also generated more numerical noise. For instance, in Case 0a, the spatial distribution of precipitation changes was notably noisy (Figure S1). Overall, convective conditions are the least ideal for winter orographic cloud seeding, as the simulated seeding impacts are either mostly negative or noisy across the target area.

The sensitivity of seeding impacts to aerosols varied by category. In Category 1, members with 10% of aerosol climatological concentration (01CCN) produced the strongest downwind shift of precipitation and therefore the weakest precipitation enhancement or even a slightly negative impact (Figure 9c). This is because the low CCN condition facilitates a more effective warm-rain process and inhibits ice-phase processes in natural conditions (see Figure 11b,c from CCN member in Case 1b and Figure 11e,f from its respective 01CCN member). Therefore, low CCN concentration resulted in a stronger seeding-induced phase transformation from liquid- to ice-phase precipitation, shifting the precipitation enhancement out of the target area ( Figure 12). In addition, the increase in ice precipitation in 01CCN members was offset by an even stronger reduction in liquid

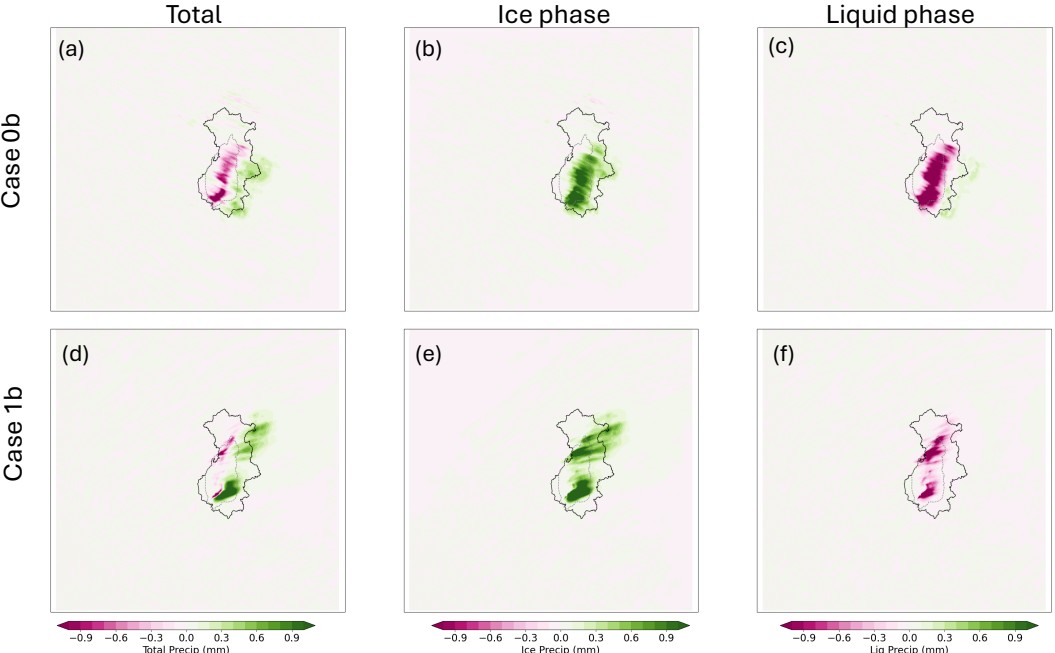

**Figure 8.** The changes in accumulated precipitation (SEED-CTRL) due to seeding for (a, d) total, (b, e) ice-phased, and (c, f) liquid-phase precipitation for Case 0b and Case 1b (see labels on the left side). The target area is outlined in grey thin line, and the Snowy water catchment is outlined in black line.

precipitation. Nevertheless, the seeding impacts are less sensitive to aerosol concentrations in Category 0 (Figure 9c), where
no obvious low/high bias was seen. This implies that aerosol impacts in glaciogenic seeding in deep, convective clouds might not be important compared to stratiform conditions.

The seeding impacts also exhibited sensitivity to the ice nucleation schemes. Compared to the DeMott scheme, the Meyers scheme, which assumes a constant INP concentration and only depends on temperature, tends to produce less negative seeding impacts in Category 0 cases, but no consistent pattern was found in Category 1 cases (Figure 9b). The Meyers scheme has
340 a much higher nucleation efficiency and creates higher ice concentrations in natural (no seed) conditions. This reduced the SLW and inhibited the already active ice precipitation processes in Category 0 cases, resulting in a suppressed liquid-phase precipitation in natural condition and a reduced ice-precipitation enhancement by seeding. Therefore, Meyer schemes generated weaker seeding-induced precipitation changes. This trend was also found in the case study by Chen et al. (2023, Figure 9 of their paper).

Large-scale forcings (initialization datasets) contribute significantly to the ensemble spread, in particular in Category 0 cases. The spread is found largely clustered by datasets (Figure 9a). For instance, in Case 0c, BARRA-driven members showed significantly high negative seeding effects (Figure 9a), and the difference between ERA5 and CFSv2 was less significant. This is because in this case BARRA-driven members produced moderately more liquid-phase precipitation and much less

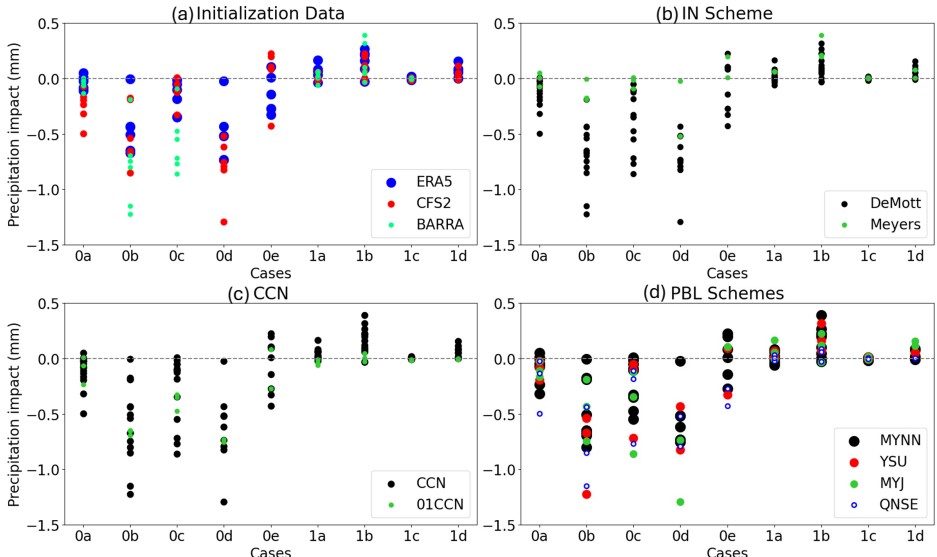

**Figure 9.** Same as Figure 3 but for the ensemble spread of the seeding impact in accumulated precipitation averaged over the target area.

ice-precipitation in the catchment in natural (no seed) conditions. This consequently caused a strong seeding-induced pre-
cipitation phase conversion from liquid to ice. Ultimately, that paired with a strong wind resulted in a reduction in total
precipitation within the target area and an increase to the east of the area. In contrast, weaker seeding effects were seen in
the ERA5 and CFSv2 members. Due to less liquid phase precipitation in the natural condition and more pre-existing natural
ice processes, less phase conversion was noted. In Case 0d, only one CFSv2 member (CFS2_CCN_DeMott_MYJ) produced
substantially more negative impact than the rest of the members, and one ERA5 member produced the least negative impact
(ERA5_CCN_Meyers_MYNN). The rest of the members within those groups shared a similar range of spread. As a result,
the precipitation pattern changes due to seeding is largely similar between the two datasets. In Case 0e where about half the
members produced negative seeding impacts and the other half positive (Figure 9a), initialization datasets were the dominant
factor in determining the sign of seeding impacts: most CFSv2 members produced positive seeding impacts, and most ERA5
members see a negative seeding impact. This demonstrates that in the high wind conditions, the initialization datasets providing
the prevailing wind fields are critical in dispersing AgI, transporting precipitating hydrometeors, and finally determining the
impacted seeding area. In addition, whether riming is efficient in seeded and unseeded clouds highly depends on the avail-
ability of the SLW, which is also mostly sensitive to the initialization datasets, as demonstrated in Section 3.1.1. Finally, the
spread caused by different PBL schemes appeared evenly distributed across the seeding impact range and no obvious common
patterns can be found across the cases.
To better quantify model uncertainty attributed to different configurations, a differential analysis on seeding-induced precipi-
tation changes was conducted, similar to Section 3.1.2, to isolate the ensemble difference arising from each model configuration
relative to the default one. Figure 10(a) shows that in Category 1 (stratiform) storms, IN schemes continue to contribute the

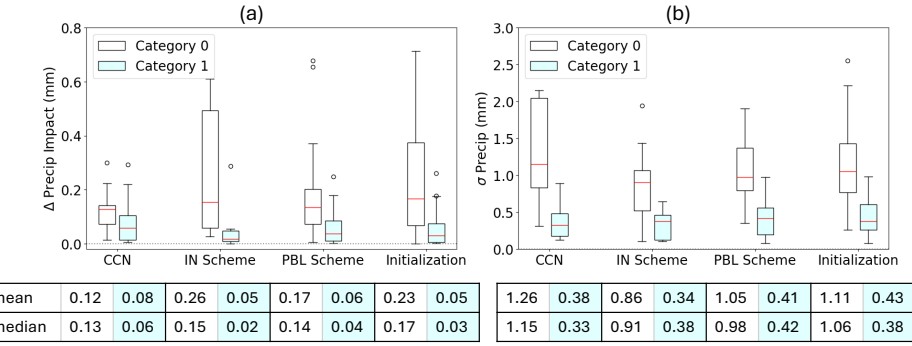

**Figure 10.** Same as Figure 5 but for the ensemble difference of the seeding impact in accumulated precipitation averaged over the 1km model domain.

least to the model spread in domain-average seeding-induced precipitation changes, while CCN and PBL schemes contribute the most. However, Category 0 (convective) storms display a different pattern: IN schemes emerge as the leading contributor to ensemble spread (mean value of 0.26 mm), followed by initialization datasets. Unlike the cases for the natural precipitation, where the model spread due to microphysics (CCN and IN scheme) shows similar ranges across storm types (Figure 5(a)), the uncertainty in seeding impacts is substantially larger for convective storms (Figure 10).

In terms of spatial variability (Figure 5(b)), CCN concentration is the leading contributor to ensemble spread in seeding response for Category 0 (mean standard deviation of 1.26 mm), followed by initialization dataset (1.11 mm), PBL scheme (1.05 mm), and IN scheme (0.86 mm). As with domain-averaged differences, spatial variability is substantially higher for Category 0 cases. In Category 1 cases, all four configurations contribute similarly to the model differences.

### 3.2.2 AgI Dispersion and Impacts on Cloud Profiles

To better understand the cloud response to seeding at a process level, vertical cross-sections along the prevailing wind direction were analyzed for each case. These cross-sections include a northern transect passing through the Cabramurra site and a southern transect intersecting the Khancoban site (Figure 13). The transects were selected based on the prevailing wind direction unique to each case, ensuring alignment with the dominant meteorological flow patterns.

The high terrain to the west of the catchment acted as a barrier for the eastward transport of AgI particles released by the ground generators on the western slope (Figure 1(b)). Additionally, less generators were deployed to the central target area compared to the north and the south. These two factors combined led to a lower AgI particle concentration in the target's center, as shown in Figure 13 for Case 0b and Case 1b and Figure S3 for all cases. Consequently, the seeding-affected region split into northern and southern sections for most of the cases, as shown in Figure 8. In addition, most of the southern ground generator sites are located at a lower elevation compared to the northern sites, and thus the released seeding particles were easier to be locally trapped in the valley, as can be seen in the highly concentrated value near the southern sites in Figure 13 and Figure S3.

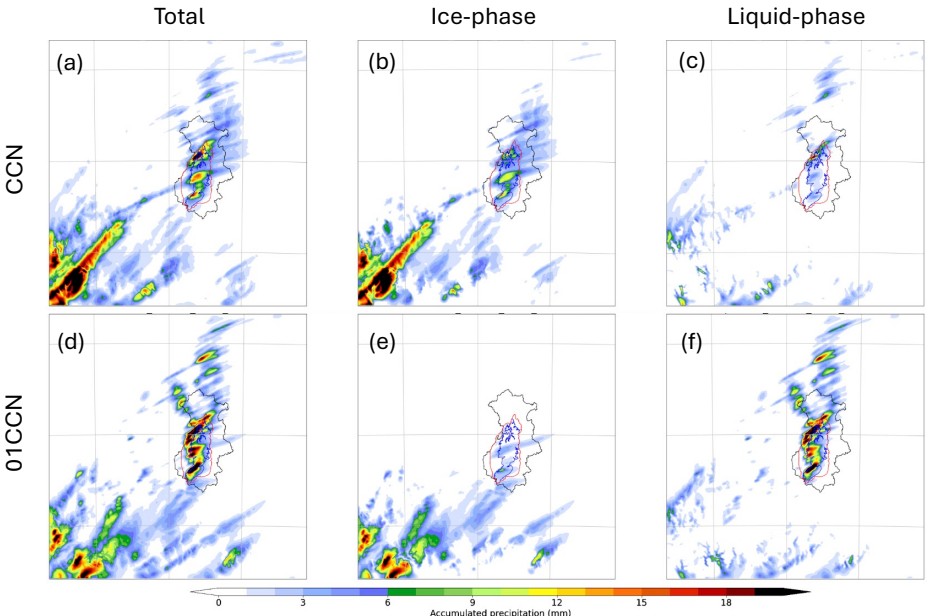

**Figure 11.** Same as Figure 8 but for the accumulated precipitation from Case 1b's Member 1 (ERA5_CCN_DeMott_MYNN, labeled as CCN) and member 5 (ERA5_01CCN_DeMott, labeled as 01CCN).

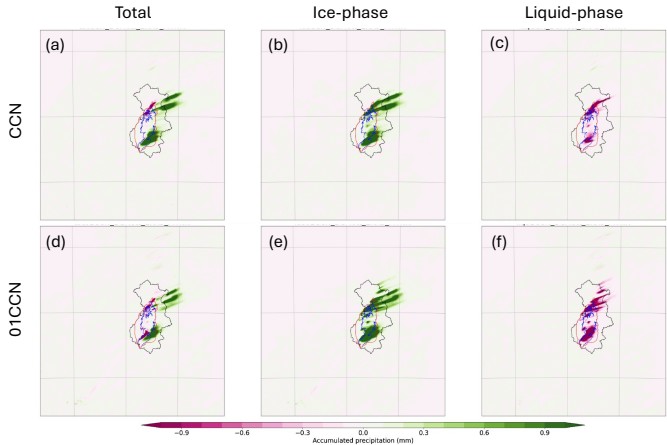

**Figure 12.** Same as Figure 11 but for the changes in accumulated precipitation due to seeding (SEED - CTRL).

SLW was primarily located on the west side of the catchment (Figure S2). Although AgI particles could travel further east in the atmosphere (Figure 13b,d), their downstream cloud impacts were minimal. This is also supported by distribution of precipitation changes in Figure 8, and most of the enhanced precipitation happening outside the catchment was attributed to phase-transformation-induced precipitation shift discussed above, rather than the direct effects of seeding material on downstream clouds.

It is also noticeable that there are clearly two distinct prevailing wind patterns between the Category 0 and 1 cases. The convective cases from Category 0 were primarily dominated by northwestern winds (e.g., Figure 13 (a) for Case 0b), and in stratiform cases from Category 1, westerly or southwesterly winds prevail (e.g., Figure 13(c) for Case 1b, see also Figure S3 for all cases). Clearly, the prevailing winds not only provide distinct large-scale forcings to the storms, but also strongly influence how the air flow interacts with the terrain and thus the turbulent-mixing and transport of cloud particles within the atmospheric boundary layer. It is noticed that negative seeding impacts are strongly associated with northwesterly winds. In Category 0, where Case 0a and Case 0e show more westerly winds, slightly less negative or even positive impacts were seen (Figure 7). Vertical cross-sections of clouds and AgI dispersion reveal details in microphysical response to seeding. The condition for which AgI particles effectively nucleate ice to enhance the precipitation within the target area depends on a few critical factors. For example whether the wind is strong enough to transport AgI particles to the optimal temperature zone ($\sim$ -7 to -15 $^\circ$C), but not too strong to blow them away from the target area before the seeding impacts occur, whether sufficient amounts of supercooled liquid are present in the clouds, and how active the natural ice process is already to generate ice-phase precipitation processes. It can be seen that even within the same storm, cloud profiles can vary significantly in time and space due to the complicated interaction between AgI, clouds, and the complex terrain. For example, Case 0b sees generally negative seeding impacts from the accumulated precipitation either over the target area or over the catchment (Figure 7). Nevertheless, enhancement in ice precipitation can be seen in the northern transect with no reduction in liquid precipitation within the catchment (Figure 14(a2)).

In addition, cloud profiles vary significantly between the northern and southern catchment. For example in Case 1b, the northern transect shows significant seeding impacts (Figure 14(c2)). The abundant SLW with cloud top temperature between -15 and -20 $^\circ$C (Figure 14(c1)) led to extensive downwind spread of nucleated ice by AgI (Figure 14(c3)). A high concentration of cloud ice formed between -10 $^\circ$C and -15 $^\circ$C, and the enhanced graupel and snow formation happened below the ice layer. Contrary to the control (no seed) simulation where rain was predominant between 40 km and 60 km distance in the cross-section (Figure 14 (c1)), the seeding completely suppressed liquid phase processes and converted cloud water and rain into ice precipitation falling in the 40-80 km distance, extending approximately 40 km further downwind. This further supports the finding that seeding transforms liquid precipitation to ice precipitation, which is transported further downwind. For the southern transect, the seeding impact is less pronounced due to the pre-existing active ice precipitation processes in the control simulation (Figure 14(d1)). Regardless of cloud ice formation on the eastern edge of the cloud near 40 km distance, this enhanced ice production does not lead to precipitation enhancement reaching the ground (Figure 14 (d2)). Overall, seeding produced ice further downwind in the northern region ((a3) and (c3) in Figure 14 and 15) compared to the southern region ( panels (b3) and (d3) in Figure 14 and 15). This resulted in a more widely spread seeding impact along wind in the north

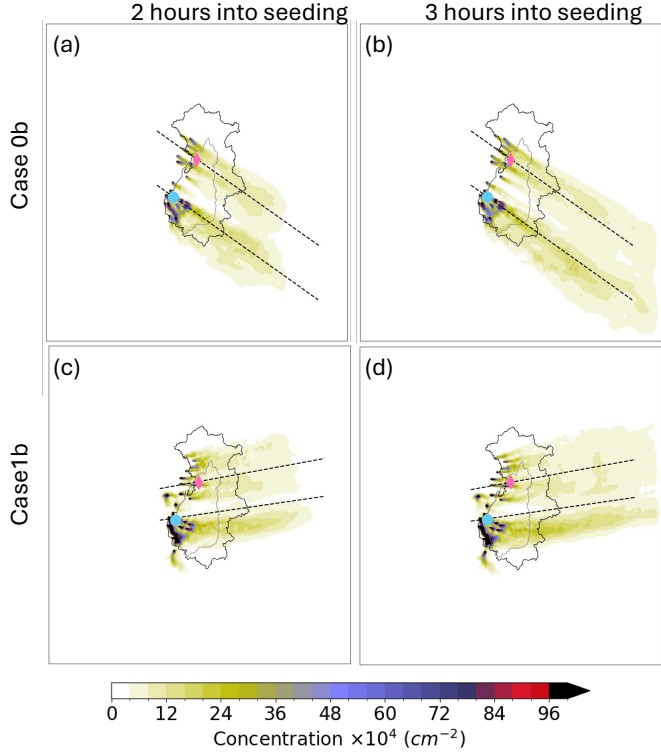

**Figure 13.** The ensemble mean of AgI dispersion over the domain from (a-b) Case 0b and (c-d) Case 1b. The value indicates vertically integrated AgI concentration $(cm^{-2})$. The left column (a,c) is from two hours after seeding starts, the right column (b,c) is from three hours after seeding starts. The dashed lines represent two transects along the prevailing wind of the cases: one from northern catchment passing through the Cabramurra site (red dot in the domain), referred to as the northern transect, and the other from southern catchment passing the Khancoban site (blue dot in the domain), referred to as the southern transect. The target area is marked in gray lines and the water catchment is in black lines.

(Figures 8 and 12). While in the southern region, the nucleated ice stays primarily within the target area. It is also evident that the cloud response to seeding and AgI transport are also sensitive to the large-scale forcing, and using different initialization datasets lead to differences in natural supercooled liquid cloud profiles, AgI transportation, and thus the changes in clouds precipitation due to seeding, as can be illustrated by the differences between ERA5-driven member results in Figure 14 and BARRA-driven member results in Figure 15. Using different PBL schemes also result in different AgI dispersion profiles and

cloud profiles (not shown here). Therefore, the ensemble approach which considers the range of uncertainties from variations in model setups is critical for a systematic evaluation of the seeding impacts, rather than relying on a single model output.

In conclusion, category 1 cases showed general precipitation increases over the target area across the ensemble members; and category 0 cases lead to reductions or relocations of precipitation that resulted in generally reduced precipitation over the target areas. The terrain also played a role in AgI dispersion. High western terrain blocked AgI transport into certain areas, leading

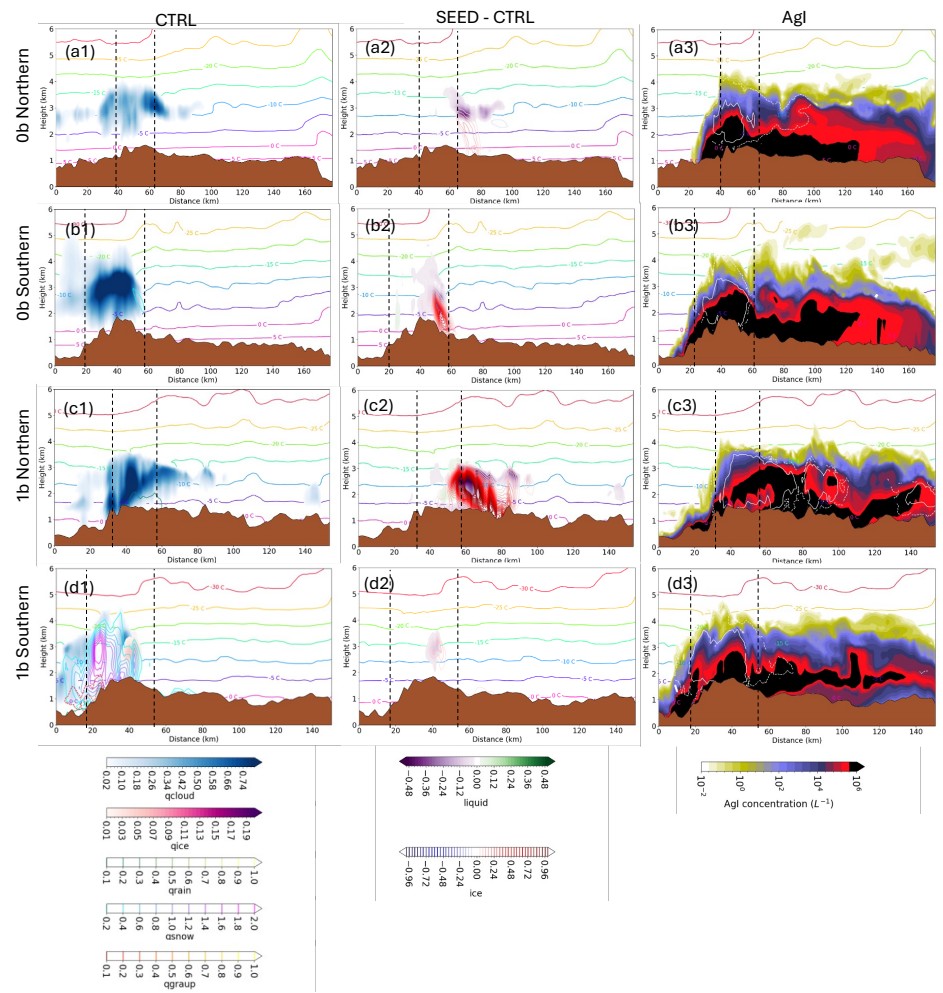

**Figure 14.** Vertical cross-section along the prevailing wind of (a1-d1) the cloud hydrometeor mixing ratio profiles, (a2-d2) the cloud hydrometeor mixing ratio changes (g kg$^{-1}$) due to seeding (SEED - CTRL), and (a3-d3) total concentration of AgI two hours after seeding starts in the ensemble member of ERA5_CCN_DeMott_MYNN (member 1 in Table 1). The vertical dashed black lines indicate the border of the target area. AgI carried by cloud ice is displayed in white contour lines, where dashed white contour indicates area with concentration of AgI carried by ice > 0.01 L$^{-1}$, and solid white contour shows area with concentration of AgI carried by ice > 100 L$^{-1}$. From top to bottom are the cross-sections from northern transact in Case 0b, southern transact in Case 0b, northern transact in Case 1b, and southern transact in Case 1b, respectively. The northern and southern transacts from Case 0b and 1b are indicated in Figure 13

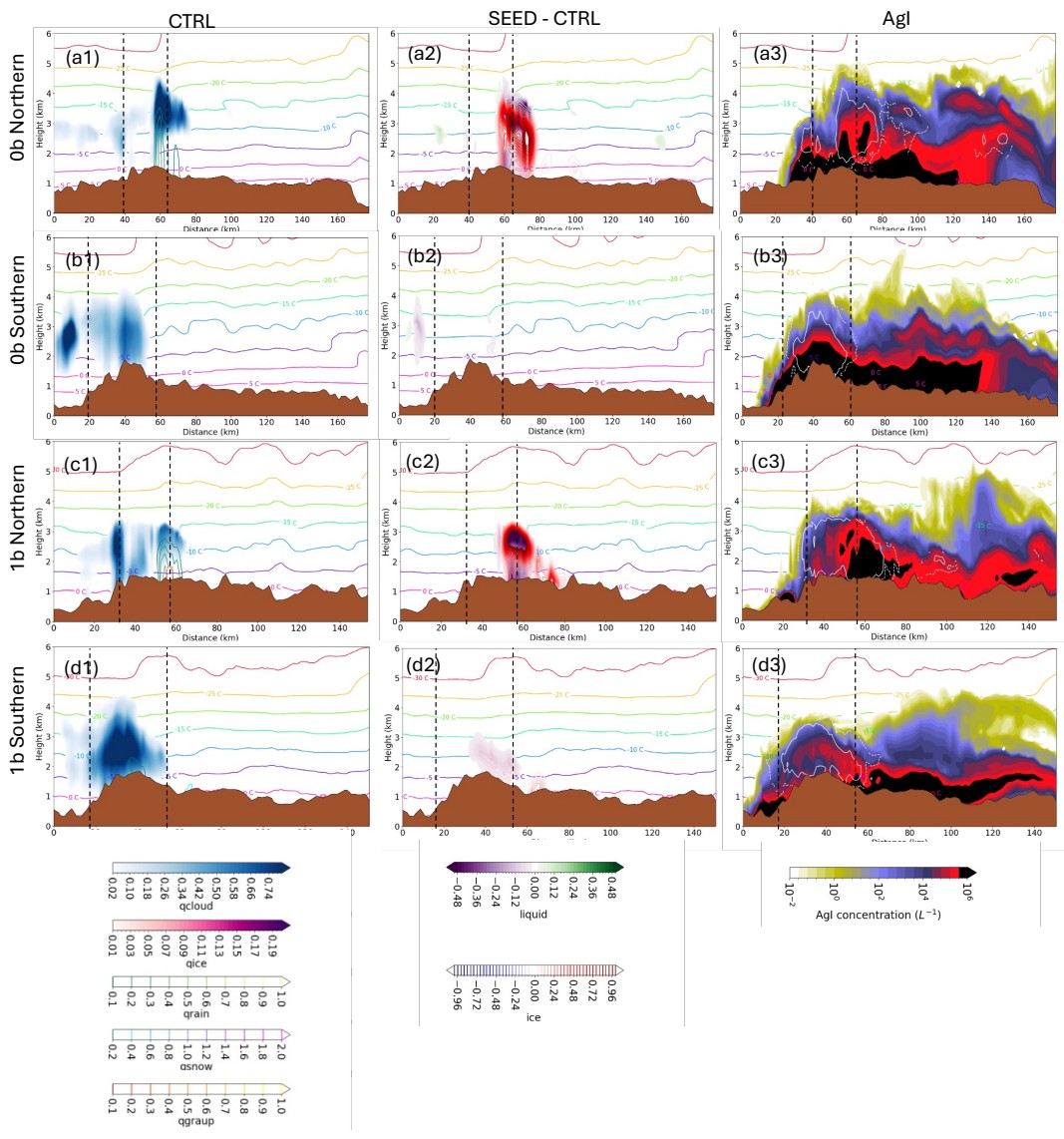

**Figure 15.** Same as Figure 14, but for the ensemble member of BARRA_CCN_DeMott_MYNN (member 13 in Table 1).

to spatial variability in seeding impacts. These process-level analyses highlight the complex interplay between meteorology, terrain, and seeding efficacy.

## 4    Conclusions and Future Work

This study builds upon previous work to systematically evaluate glaciogenic cloud seeding impacts over the Snowy Mountains using an ensemble modeling framework. By incorporating 18 ensemble members with varied model configurations and initial-

ization datasets, including the novel use of the BARRA dataset, this research investigates seeding impacts under two dominant wintertime meteorological regimes.

The analysis highlights the value of including multiple initialization datasets to capture the range of model uncertainties. Among the datasets tested, BARRA-driven simulations generally outperform ERA5 and CFSv2 in representing SLW. This suggests that BARRA is a reliable reanalysis dataset for driving high-resolution WRF simulations in southeastern Australia,

which also shows potential for retrospective numerical simulations for other regions in Australia, New Zealand, and SE Asia in the future.

The results confirm that both model sensitivity is highly case-dependent and storm-type-dependent. PBL schemes and initialization datasets are dominant sources of uncertainty in domain-averaged natural precipitation for both convective and stratiform storms, and IN scheme shows least model sensitivity. Quite opposite, seeding-induced precipitation changes show greatest sen-

sitive to IN schemes, in particular in convective cases. This quantified analysis in natural precipitation and seeding impact imply that while initialization datasets primarily determine the background cloud and precipitation structures, seeding-induced changes are highly sensitive to both microphysical uncertainties and PBL processes. Natural CCN and IN processes could influence the prevalence of SLW and natural ice, which in turn affect AgI nucleation efficiency, competition of SLW, and the overall seeding response. PBL processes also influence how seeding material disperse vertically and horizontally within the boundary

layers, which influence seeding impacted area and precipitation changes. Finally, the consistent greater ensemble spread in Category 0 storms emphasizes the challenges of simulating and evaluating seeding outcomes in convective environments.

Importantly, the dominant contributor to model uncertainty not only varies by storm type but also varies across cases within each storm type. For example, initialization datasets contribute to the largest model uncertainties in Cases 0a and 0e, while changing aerosol concentration leads to the largest ensemble spread in Cases 0b, 0c, 1a, and 1c, changing aerosol concentrations

and 01CCN concentration members produced the highest precipitation in those cases. However, 01CCN members in other cases do not show this high bias. In Cases 0d, 1b, and 1c, PBL schemes are the dominant contributor to the ensemble spread. This implies that the leading uncertainty of the model can differ in different meteorological conditions within each storm type, indicating the importance to use ensemble approaches to appropriately account for the range of uncertainty.

Comparison between the seeded and control (no seed) simulations indicates that seeding impacts vary significantly among

the members and across different cases, with clear, distinctive patterns in the two storm categories. Category 0 cases, with deep, convective clouds and active precipitation processes, are considered the least ideal for cloud seeding, with four out of five cases showing reduction in mean accumulated precipitation over the target area. The high wind condition in Category 0 cases

facilitates the downwind transportation of the seeding-converted, slow-falling ice-phase hydrometeors, leading to precipitation enhancement outside the target area. It was also found that seeding inhibited the already efficient ice-precipitation processes, likely with strong riming, in those cases by increasing ice concentration and reducing the rimed water. Category 1 cases, which feature stable, stratiform conditions and weak precipitation, show seeding enhancement in simulated mean precipitation over the target area for all cases. It was also noted that an extremely clean environment (10% of aerosol climatology) could hamper the precipitation enhancement. This is because pristine environments result in less supercooled droplet concentrations and more efficient removal of supercooled liquid through warm-rain processes in natural conditions.

Regardless of the opposite seeding signals from the two categories, the high degree of variability between ensemble members in all cases except Case 1c indicates that simulated cloud seeding response is highly sensitive to both large-scale forcing and model configurations. Ensemble analysis indicates that there is no single model configuration that optimally represents all cases/categories. Even within the same meteorological regime, opposite model biases can be found in different cases using the same ensemble configuration due to the complex physical processes involved. Therefore, interpreting model results from a single model configuration should be done with caution. This supports the need to use ensemble approaches for assessing the cloud seeding impacts.

Previous evaluations of the cloud seeding impact in the Snowy Mountains have demonstrated an overall positive and statistically significant seeding impact based on traditional statistical analysis using precipitation observations from a network of precipitation gauges and multi-year randomised seeding statistics (Manton and Warren, 2011; Manton et al., 2017; Smith et al., 1963; Smith, 1967). The current study supports findings from previous statistical analysis which also show significant variability in seeding impacts, for example, Smith et al. (1963) shows that the seeding impacts over Snowy Mountains varied significantly from year to year. Our numerical studies indicate that such variability highly depends on differences in meteorological factors, such as cloud types and prevailing winds associated with individual weather events. This analysis complements previous efforts by providing storm-specific seeding impacts and variability, which will ultimately support the identification of favorable seeding conditions in a weather modeling framework.

However, the differences between these statistical and simulation assessment methods pose challenges in conducting consistent and comprehensive comparisons of the seeding impacts. To address this, conducting seasonal or even multi-year ensemble simulations that include all seeding cases during the study period will be essential for accurately quantifying overall seeding impacts. Additionally, developing a comprehensive storm climatology (including storm categories, frequency, and trends) over the region will further aid in assessing and predicting the climatology of seeding impacts. In addition, detailed investigations will be needed to quantify the sensitivity to certain microphysical processes for better constraining the ensemble spread, as well as better understanding the seeding mechanism in the complex mixed-phase cloud environment. For example, how seeding impacts are modulated by the riming efficiency used in the microphysical scheme in different weather conditions and thus determine the fraction of rimed and unrimed snow and graupel in the precipitation. This will be critical because riming significantly affects the hydrometeor's density and fall speed and therefore the transport and relocation of precipitation.

*Data availability.* The model output data is stored in the Amazon Web Service (AWS) S3 storage. Due to the large size of the ensemble simulation dataset, the data will be available only upon request.

*Author contributions.* SC and LX contributed to the development of the methodology and determination of the ensemble members. TC, AP,
SK, and JP prepared and quality controlled the observational datasets. SC ran the simulation, conducted the data analysis, created figures,
transferred data to AWS S3, and wrote the initial draft of the manuscript. ST and TC provided guidance and oversight for the project. JW
was responsible for managing and coordinating the research project. BP set up the AWS cloud platform and tested the WRF-WxMod at the
initial phase and provided technical support during the ensemble simulation. All authors contributed to the discussion and interpretation of
results and to the review and editing of the manuscript.

*Competing interests.* The authors declare no competing interests

*Acknowledgements.* This research was supported by Snowy Hydro Ltd. This material is partially based upon work supported by the NSF
National Center for Atmospheric Research (NCAR), which is a major facility sponsored by the U.S. National Science Foundation under
Cooperative Agreement No. 1852977.

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
