# Peer review of "Assessing glaciogenic seeding impacts in Australia's Snowy Mountains: an ensemble modeling approach"

_EGUsphere, 2025_

## Referee Comment (RC1)

**Review of "Assessing glaciogenic seeding impacts in Australia's Snowy Mountains: an ensemble modeling approach" by Chen et al. (egusphere-2025-1434)**

This study uses ensemble numerical simulations to assess glaciogenic cloud seeding over Australia's Snowy Mountains. The study convincingly shows that certain meteorological conditions benefit seeding while others do not. Moreover, the impact of further physical parameters (CCN concentration) and modeling choices (boundary layer scheme, ice nucleation scheme, large-scale drivers/initial conditions) is addressed. While I generally agree with the simulation results presented, I miss a more quantitative assessment of the impact modeling choices can have, as I will detail below.

**Major Comments**

*Quantifying Differences.* The study shows nicely how physical parameters and modeling choices affect the results (e.g., Figs. 3 and 8). However, I am wondering how to interpret the presented differences. I believe that the use of different large-scale drivers (ERA5, CFS2, BARRA) is a step in the right direction, as they represent "perturbed" initial states. I suggest creating initial data sets that are systematically perturbed to represent the natural variability expected in this region. This will enable the reader to assess if differences in the boundary layer scheme, ice nucleation scheme, or CCN concentration are significant, i.e., visible beyond the natural variability.

**Minor Comments**

Ll. 43 – 47: Is there a reasonable argument to assume that cloud microphysical processes and their parameterization are different for the Northern and Southern hemispheres?

L. 91: This should be the geometric mean diameter.

L. 91: State diameter in microns?

L. 93: Why only turbulent collection?

Ll. 99 – 102: 20.6 g/h seems to be very small. For a particle diameter of 40 μm, this results in an injection rate of roughly $10^8$ particles per hour, which is not a lot considering the dispersion of the plume.

Sec. 2.2.1: Is the model also nudged to the large-scale conditions?

L. 150: A "range of CCN concentrations" is not analyzed. Only one smaller concentration is addressed.

Ll. 160 – 162: How did the authors consider the extent of uncertainties in constructing their ensemble? The outlined approach feels very subjective.

Tab. 1: Highlight the changed parameters, e.g., by bold characters.

L. 236: How do you determine how much the aerosol concentration contributes to the spread?

Fig. 6, ll. 272 – 273: Why use gigalitres, not mm?

Figs. 12 and 13: The contours for qice, qsnow, qgraup, and ice are very hard to distinguish from each other and the temperature contours in the background.

**Technical Comments**

L. 8: Define BARRA.

L. 9: Define PBL.

Ll. 18 ff.: SLW has already been defined.

L. 52: Define LWP.

Ll. 91 ff.: Use non-italic characters for units.

Ll. 160 – 170: This paragraph should become its own subsection (2.2.5).

---

## Author Comment (AC1)

Response letter

We appreciate both reviewers' time and efforts in providing the careful and constructive comments that help improve the quality of our manuscript. We have created a point-by-point response below with the reviewer's comments in black and our response in blue.

**Reviewer #1**
This study uses ensemble numerical simulations to assess glaciogenic cloud seeding over Australia's Snowy Mountains. The study convincingly shows that certain meteorological conditions benefit seeding while others do not. Moreover, the impact of further physical parameters (CCN concentration) and modeling choices (boundary layer scheme, ice nucleation scheme, large-scale drivers/initial conditions) is addressed. While I generally agree with the simulation results presented, I miss a more quantitative assessment of the impact modeling choices can have, as I will detail below.

Major Comments

*Quantifying Differences.* The study shows nicely how physical parameters and modeling choices affect the results (e.g., Figs. 3 and 8). However, I am wondering how to interpret the presented differences. I believe that the use of different large-scale drivers (ERA5, CFS2, BARRA) is a step in the right direction, as they represent "perturbed" initial states. I suggest creating initial data sets that are systematically perturbed to represent the natural variability expected in this region. This will enable the reader to assess if differences in the boundary layer scheme, ice nucleation scheme, or CCN concentration are significant, i.e., visible beyond the natural variability.

We thank the reviewer for this important point regarding quantifying and differentiating the ensemble spread attributed to different modeling choices relative to natural variability.

In the original submission, Figure 2 provides a qualitative comparison of the ensemble variability (as shaded areas) with natural variability (thick black curve from observations) over the region. The differences among shaded areas of different colors reflect the sensitivity to initialization dataset, while the shaded areas represent the ensemble spread within each initialization dataset. A similar comparison was shown in Figure 5 using sounding.

Systematically isolating and quantifying the variance contribution of each modeling choice is not straightforward in our ensemble design, as we strategically did not run all exhaustive ensemble members (i.e. not all combinations of initialization × CCN × INP × PBL are present) due to computational constraints.

To address this concern, we developed a structured approach using differential comparisons relative to a fixed default ensemble member. The default configuration is defined as "CCN_DeMott_MYNN", present under each initialization dataset group. The variation members always have only one changed parameter (CCN, IN scheme, or PBL scheme) relative to the

default member, across the three initialization datasets (as highlighted in bold in revised Table 1). Therefore, in each case, we have:

- 6 members for CCN sensitivity (2 CCN configurations × 3 initialization datasets)
- 6 members for IN sensitivity
- 12 for PBL (4 PBL configurations × 3 initialization datasets)
- As all initialization datasets contain 6 equal members, which gives us in total 18 members (= 3 initialization datasets * 6 members).

This allows ensemble difference statistics to be compared across all modeling variations on an equal footing.

Figure A below shows boxplots of (a) domain-averaged and (b) spatial standard deviation of accumulated precipitation across all ensemble members grouped by parameter variation. We stratify results by storm type: Category 0 (convective) and Category 1 (stratiform). It is found that PBL scheme and Initialization dataset variations produce the largest domain-average precipitation differences, particularly for Category 0 cases. The mean ensemble difference attributed to PBL is 0.29 mm in Category 0 vs 0.16 mm in Category 1. For initialization datasets, the mean difference is 0.29 mm and 0.22 mm, respectively. In contrast, changes in CCN and IN schemes yield smaller differences and are insensitive to storm category (Figure A(a)).

In terms of spatial variability of the ensemble differences (Figure A(b)), all model perturbation groups show greater sensitivity in Category 0 cases, reflecting the inherently higher uncertainty in convective systems. Similar to the domain-averaged differences, initialization and PBL produce the largest spatial spread, with a mean standard deviation of 2.97 mm and 2.75 mm in Category 0, respectively, vs 1.76 mm and 1.67 mm in Category 1. Interestingly, CCN variations also caused substantial spatial spread (mean standard deviation value of 2.59 mm in Category 0 and 1.77 mm in Category 1), nearly comparable to PBL schemes and initialization dataset groups. IN schemes contribute the least spread.

[Figure]

| | | | | | | | | | | | | | | | | |
|---|---|---|---|---|---|---|---|---|---|---|---|---|---|---|---|---|
| mean | 0.18 | 0.17 | 0.11 | 0.11 | 0.29 | 0.16 | 0.29 | 0.22 | 2.59 | 1.77 | 1.55 | 1.15 | 2.75 | 1.67 | 2.97 | 1.76 |
| median | 0.17 | 0.16 | 0.10 | 0.08 | 0.19 | 0.13 | 0.26 | 0.16 | 2.52 | 1.76 | 1.88 | 0.86 | 2.74 | 1.71 | 2.69 | 1.57 |

Figure A: (a) Domain-average and (b) standard deviation of the spatial distribution of ensemble precipitation differences (in mm) over the 1 km domain. Ensemble differences attributed to variations in CCN concentration, IN schemes, PBL schemes, and initialization datasets are shown separately. Results for Category 0 (white) and Category 1 (light cyan) cases are also

displayed side by side within each variation. The domain-averaged values used for (a) were absolute values to avoid crossing zero in comparison. The mean value and median value for storm category are shown in the table beneath each variation group.

As for seeding impact uncertainties, results show stronger model sensitivities for Category 0 cases (Figure B), which again indicates that simulating and quantifying seeding impacts in convective cases is with higher uncertainty than in stratiform cases. The model uncertainties attributed by the four model variations in Category 1 cases are similarly low. For Category 0 cases, IN (mean value of 0.26 mm) and Initialization datasets (mean value of 0.23 mm) dominate the model uncertainty in simulating the area-averaged seeding impacts (Figure B(a)). PBL schemes (0.17 mm) and CCN (0.12 mm) have smaller contributions. This pattern contrasts with the total accumulated precipitation in Figure A(a), where PBL schemes and initialization datasets dominate. For spatial variability of the model uncertainty, CCN emerges as the leading contributor, followed by initialization dataset, PBL scheme, and IN schemes (Figure B(b)).

[Figure]

| | CCN | | IN Scheme | | PBL Scheme | | Initialization | | CCN | | IN Scheme | | PBL Scheme | | Initialization | |
|---|---|---|---|---|---|---|---|---|---|---|---|---|---|---|---|---|
| mean | 0.12 | 0.08 | 0.26 | 0.05 | 0.17 | 0.06 | 0.23 | 0.05 | 1.26 | 0.38 | 0.86 | 0.34 | 1.05 | 0.41 | 1.11 | 0.43 |
| median | 0.13 | 0.06 | 0.15 | 0.02 | 0.14 | 0.04 | 0.17 | 0.03 | 1.15 | 0.33 | 0.91 | 0.38 | 0.98 | 0.42 | 1.06 | 0.38 |

Figure B: same as Figure A but for ensemble difference in seeding-induced precipitation changes over the target area(mm).

These results verify that models are more sensitive to configurations in simulating convective conditions. In general, PBL schemes and initialization datasets are the dominant factor in model uncertainty in simulating the natural precipitation process. For seeding impact, the dominant source of model uncertainties depends on the storm type. And microphysics (CCN and IN) may become more important.

We have incorporated above expanded quantitative analysis in Section 3.1.2, Section 3.2.1, and Conclusions and Future Work in the manuscript and also included Figures A ( Figure 5 in revision) and Figure B (Figure 10 in revision) to support this discussion.

Minor Comments

Ll. 43 – 47: Is there a reasonable argument to assume that cloud microphysical processes and their parameterization are different for the Northern and Southern hemispheres?

It is not that the parameterizations themselves should be different, but rather that their calibration and performance may vary across regions/hemispheres due to differing ambient conditions, and thus warrant careful evaluation.

While the fundamental microphysical processes themselves do not differ by hemisphere, ambient aerosol and INP concentrations — which strongly influence cloud microphysics — do show clear hemispheric differences. For example, the Southern Hemisphere is known to be more pristine, with substantially lower CCN and INP concentrations due to limited anthropogenic and terrestrial sources. This affects cloud phase partitioning, riming processes, and potentially the interactions between the seeding particles and clouds. And most microphysics parameterization schemes were tuned and tested in the Northern Hemisphere environments (e.g. Thompson et al. 2008, 2014; Eidhammer et al. 2009; Meyers et al. 1992),  which may (or may not) show differences in the model performance for the Southern Hemisphere. We modified the statements and included the above-mentioned citations in the introduction for further clarification:

*"However, the unique conditions in the Snowy Mountains also introduce potential challenges for modelling precipitation processes and the seeding impact compared to Northern Hemisphere environments. For example, there are uncertainties in respect to cloud phase-partitioning, riming, and collision-coalescence processes with lower atmospheric pollution, and most cloud microphysical parameterizations are developed and validated based on the observations made in the Northern Hemisphere (e.g., Meyers et al. 1992,Thompson et al. 2008 and 2014, Eidhammer et al. 2009}. These differences may affect model performance on representing clouds and precipitation processes in the Snowy Mountains. Addressing these uncertainties and challenges requires detailed numerical modeling to simulate cloud seeding under various atmospheric conditions and systematic and comprehensive assessment of the impacts.*

L. 91: This should be the geometric mean diameter.
L. 91: State diameter in microns?

You are correct, the diameter should be 0.04 $\mu m$ instead of 0.04 mm. We updated it to "*with a geometric mean diameter of 40 nm*" in the manuscript.

L. 93: Why only turbulent collection?

Thanks for pointing this out. We meant to say that AgI particles can be scavenged by the collection of liquid drops and ice crystals through Brownian diffusion, turbulent diffusion, and phoretic effects (thermophoresis and diffusiophoresis). Details of the AgI scavenging formulation is based on Caro et al. (2004) and can be found in Xue et al. (2013) which uses the formulation from Caro et al. (2004) to determine the fraction of AgI scavenged.
We have modified the statement to more correctly describe the AgI scavenging and activation process:

*"AgI particles are treated as a single-mode lognormal distribution (with a geometric mean diameter of 40 nm and a geometric standard deviation of 2) and can act as INPs or cloud condensation nuclei (CCN) due to their soluble components. The AgI particles can be scavenged by liquid drops and ice crystals through processes such as Brownian diffusion, turbulent diffusion, and phoretic effects (thermophoresis and diffusiophoresis), which subsequently determines the AgI nucleation through immersion freezing and contact freezing. The detailed description and formulation can be found in Xue et al. (2013a)".*

Ll. 99 – 102: 20.6 g/h seems to be very small. For a particle diameter of 40 µm, this results in an injection rate of roughly 108 particles per hour, which is not a lot considering the dispersion of the plume.
Given the correct particle diameter of 40 nm, 20g/h should give us 10^9 times higher particle numbers, which is reasonable. Apologies again for the incorrect unit.

Sec. 2.2.1: Is the model also nudged to the large-scale conditions?
We did not implement additional nudging, the three reanalysis datasets here only serve as initial and boundary conditions to drive 4 km simulation which then serve as the forcing for the nested 1 km run.

L. 150: A "range of CCN concentrations" is not analyzed. Only one smaller concentration is addressed.
We have changed it to "*... two CCN concentrations (monthly climatology and 10% of climatology) were considered*"

Ll. 160 – 162: How did the authors consider the extent of uncertainties in constructing their ensemble? The outlined approach feels very subjective. Tab. 1: Highlight the changed parameters, e.g., by bold characters.

It is indeed not an exhaustive list of ensemble members, given the limited computational resources. Therefore, we aim to include the key configurations that have the potential to significantly contribute to the ensemble spread and reduce the members that did not show strong model sensitivities. In Chen et al. (2023), we found that initialization datasets contributed to the largest model spread, therefore, we added one more dataset (BARRA) that is available for the Australia region. We also want to ensure each initialization dataset group includes the same set of model variations (i.e. equal number of ensemble members under each group).

Chen et al. (2023)'s case study also ruled out some model variations that show weaker sensitivities (such as model spin-up time, Hallet-Mossop ice production efficiency). During our initial design of the ensemble members, we also ran one SKEBS (stochastic kinetic-energy backscatter scheme, Berner et al., 2011) run and one SPPT (stochastically perturbed physical tendencies, Berner et al., 2015) run to test the contribution of perturbation in initial thermodynamic fields to ensemble spread. And the results show that they only slightly altered the precipitation fields.

We included variations in the CCN, PBL schemes, and IN schemes mainly because they are critical to the cloud microphysical representation (CCN and IN) as well as the dispersion of AgI particles (PBL schemes).

To improve clarity, we have highlighted in the table the changed parameters (CCN, IN, and PBL schemes) in bold within each initialization dataset group, relative to the "default configuration" (CCN_DeMott_MYNN). The following sentence was added to the caption: "*The changed parameters are highlighted in bold within each initialization dataset group, relative to the "default configuration" (CCN_DeMott_MYNN).*"

L. 236: How do you determine how much the aerosol concentration contributes to the spread?

In the sensitivity experiments from Chen et al. (2023), we found that the aerosol climatology produced a CCN concentration with peak local value exceeding 1000 cm−3 within the simulated domain and a median concentration of around 500 cm−3. Concurrently, the cloud droplet concentration reached up to 800 cm−3 with a median value of 200 cm−3. Unfortunately, there were no in situ measurements available over the region for a quantitative verification. However, satellite retrievals of aerosol concentrations during winter months in the region (Choudhury and Tesche 2023; Yang et al. 2021; see also the online supplemental material for a detailed discussion), along with cloud droplet concentrations observed in nearby areas such as Tasmania and the Southern Ocean (e.g., Huang et al. 2021; Wang et al. 2020), suggested relatively clean conditions (on the order of 10-100 cm−3). And 10% climatology was found to produce droplet number concentration close to the observation over Tasmania and Southern Ocean. Accordingly, we adjusted the default CCN concentration to 10% of its original value, providing a lower-bound value of the aerosol concentration.

Fig. 6, ll. 272 – 273: Why use gigalitres, not mm?

It is a common practice in Australia cloud seeding programs to use gigalitres, so we keep this unit. We decided to use volume of water instead of depth (mm) because we are comparing different areas in the figure. To make it more accessible to the reader, we added in the caption:"*Box plots showing ensemble spread of the simulated precipitation changes in volume (in gigalitre (Gl), $1\,Gl\,=\,1\times10^{6}\,m^{3}$) due to seeding…*"

Figs. 12 and 13: The contours for qice, qsnow, qgraup, and ice are very hard to distinguish from each other and the temperature contours in the background.

Thanks for the comment. It is very challenging to find appropriate colormaps when we are trying to include so many variables in the same figure. But maybe the figure layout and colormaps are confusing. In fact, the colormap of qice, qsnow, qgraup is only for the left panel. While ice and liquid is for the middle panel.
In the left panel, we use filled contours for cloud hydrometeors (qice and qcloud) and line contours for the precipitating hydrometeors (qrain, qsnow, qgraup) to distinguish.

In the middle panel, filled contours for liquid phase (qcloud+qrain) and line contour for ice phase (qice+qsnow+qgraup).

To make it clear to readers that those colormaps serve for different panels, we now use boxes to separate the three panels.

Technical Comments
L. 8: Define BARRA.
Added the full name
L. 9: Define PBL.
Added
Ll. 18 ff.: SLW has already been defined.
Removed  the full spelling.
L. 52: Define LWP.
Added "*liquid water path*"
Ll. 91 ff.: Use non-italic characters for units.
Changed all units to non-italic font.
Ll. 160 – 170: This paragraph should become its own subsection (2.2.5).
Added a subsection title "*Summary of ensemble configurations*"

Reference:
Eidhammer, T., P. J. DeMott, and S. M. Kreidenweis (2009), A comparison of heterogeneous ice nucleation parameterizations using a parcel model framework, J. Geophys. Res., 114, D06202, doi:10.1029/2008JD011095.

Thompson, G., P. R. Field, R. M. Rasmussen, and W. D. Hall, 2008: Explicit forecasts of winter precipitation using an improved bulk microphysics scheme. Part II: Implementation of a new snow parameterization. Mon. Wea. Rev., 136, 5095–5115, doi:10.1175/2008MWR2387.1.

Thompson, G., and T. Eidhammer, 2014: A Study of Aerosol Impacts on Clouds and Precipitation Development in a Large Winter Cyclone. J. Atmos. Sci., 71, 3636–3658, https://doi-org.cuucar.idm.oclc.org/10.1175/JAS-D-13-0305.1.

Meyers, M. P., P. J. DeMott, and W. R. Cotton, 1992: New Primary Ice-Nucleation Parameterizations in an Explicit Cloud Model. J. Appl. Meteor. Climatol., 31, 708–721, https://doi-org.cuucar.idm.oclc.org/10.1175/1520-0450(1992)031<0708:NPINPI>2.0.CO;2.

Reviewer #2
The manuscript examines nine seeding cases using an ensemble modeling approach. The ensemble is composed of 18 members using different large-scale forcing data, planetary boundary layer schemes, ice nucleation and aerosol schemes. The nine cases investigated are divided into those with convective and stratiform precipitation. The authors found that

depending on the precipitation regime, the seeding impacted the precipitation differently. The manuscript is novel, well-structured and well-written. I have no major comments. Minor comments are given below.

We thank the reviewer for their positive comments!

Minor comments:

1. Line 8: define BARRA

Added the full name.

2. Line 9: define PBL

Added the full spelling.

3. Line 19: the degree Celsius symbol should not be in italics.

Changed all unit symbols to non-italics.

4. Line 130: It is confusing to have a sentence starting with "one of our goal…" in the ensemble design. Is this the goal of the study? The authors should rephrase the sentence for clarity.

We have modified the sentence to the following to the enhance the clarity:
"*This is the first time to our knowledge that BARRA has been used to drive WRF simulations. As mentioned in the introduction, one of our scientific objectives is to demonstrate whether BARRA can serve as a …*"

5. Line 154: Add a space between ")" and "to".

Added.

6. Line 247: The authors describe the figures in the text, which is not necessary. For example, the authors can delete "see color circles" because it is already in the figure caption. The authors should correct throughout the manuscript.

Thanks for the suggestion. We have removed such descriptions  throughout the manuscript to avoid redundancy.

7. Line 272: The units "gigalitres" is not needed since that it appears in the figure caption.

Removed.

8. Lines 297-321: This paragraph is long and is mixing results and discussion. The authors should divide it in a few paragraphs, based on the topic. For example, lines 297-300 read more like a discussion about the processes associated with the downwind shift while the sentence starting with "Furthermore" on line 300 seemed to be some results another numerical noise.

Thanks for the comment. We originally had three paragraphs but latex did not recognize it due to lacking double enters. We now have corrected it (as well as other parts that suffered the same issue).

9. Lines 347-348: The first sentence is incomplete and unclear and should be improved.

To improve the clarity, we modified the first few sentences as below:

*"The high terrain to the west of the catchment acted as a barrier for the eastward transport of AgI particles released by the ground generators on the western slope (Figure 1(b)). Additionally, less generators were deployed to the central target area compared to the north and the south. These two factors combined led to a lower AgI particle concentration in the target's center, as shown in Figure 11 for Case 0b and Case 1b and Figure S3 for all cases. "*

10. Line 377: In the sentence starting with "The abundant supercooled liquid…", "supercooled liquid" should be replaced by the acronym "SLW".

Replaced with SLW

11. Line 423: What does "negative seeding" mean?

"Negative seeding impacts" refers to the precipitation reduction due to seeding. For clarification, we modified to the following:

*"...with four out of five cases showing reduction in mean accumulated precipitation over the target area."*

12. Line 426: Should "rimed water" be replaced by "SLW"?

We modified it to *"...reduced available SLW for riming"* to better clarity.

13. Line 440: Which numerical study are you referring to? It should clarify.

This numerical study means the current study. I have modified it to "*The current study supports…*" to reduce confusion.

14. Conclusion: The authors should divide clearly the conclusion of the study and the future work. For example, the last paragraph (Line 452) reads as future work and should probably be combined with the previous one.

We divided the last two paragraphs in a way that the future work is separated from conclusions. Accordingly, the title was changed to *"Conclusions and Future Work".*